# STRUCTURE-AWARE PARAMETER-EFFICIENT MACHINE UNLEARNING ON TRANSFORMER MODELS

## ABSTRACT

Transformer has become fundamental to a vast series of pretrained large models that have achieved remarkable success across diverse applications. Machine unlearning is an emerging field focused on efficiently removing the influence of specific data from trained models, to comply with privacy regulations enforcing the right to be forgotten. The sheer size of Transformer-based models poses a significant challenge to unlearning efficiency. Existing methods find it promising to restrict unlearning updates to a small portion of influence-critical parameters. However, their parameter-efficient unlearning methods are largely devised in a structure-oblivious manner, which tends to inaccurately identify these parameters and leads to inferior unlearning performance for Transformers. In this paper, we propose `SPE-Unlearn`, a structure-aware parameter-efficient machine unlearning approach tailored for the Transformer architecture. `SPE-Unlearn` introduces a learnable pair of masks to respectively pinpoint influence-critical parameters in the heads and filters of Transformers. The learning objective of these masks is derived by jointly considering both desiderata of unlearning, i.e., sufficiency in influence removal and efficiency, and optimized through an efficient algorithm featured by a greedy search with a warm start. Equipped with the identified key parameters, `SPE-Unlearn` facilitates second-order unlearning, memory-free unlearning, and memory-aided unlearning scenarios. Extensive experiments on various Transformer models and datasets demonstrate the effectiveness and efficiency of `SPE-Unlearn` for Transformer unlearning.

## 1 INTRODUCTION

Transformer architecture (Vaswani et al. (2017)) has demonstrated superior performance in the field of natural language processing. Its models, e.g., BERT (Devlin et al. (2018)) and GPT (Achiam et al. (2023)), show impressive performance in a wide range of downstream tasks (Wei et al. (2021); Hao et al. (2019)). In light of privacy regulations, such as General Data Protection Regulation (GDPR) (Hoofnagle et al. (2019)), users are granted the right to request the removal of specific training data from models. To fulfill this requirement, machine unlearning techniques have been extensively researched (Bourtoule et al. (2021); Yao et al. (2023)). However, when applying these techniques to Transformers, which commonly involves a large number of parameters, a significant challenge lies in achieving computational efficiency while ensuring effective unlearning and preserving model fidelity (Warnecke et al. (2021); Liu et al. (2024a)).

Recent researches propose parameter-efficient unlearning techniques (Liu et al. (2024a); Pochinkov & Schoots (2024); Schoepf et al. (2024)), which identify the influence-critical parameters to govern the unlearning process. Specifically, these methods assess the importance of parameters through different evaluation strategies, allowing selective updates to reduce computational overhead and improve unlearning efficiency. However, applying parameter-efficient unlearning to address the dilemma of the unlearning tasks in Transformers faces two major limitations. First, previous evaluation methods rely on heuristic or empirical strategies to identify parameters. For Transformer models with an immense number of parameters, identifying those specifically relevant to unlearning becomes inefficient. Additionally, existing methods (Pochinkov & Schoots (2024); Liu et al. (2023b); Shi et al. (2023)) assess importance of parameters by comparing performance (e.g., activations) on forgetting dataset and remaining dataset may result in sub-optimal selection process for unlearning. Second, previous unlearning methods overlook the intricate interactions between

structures in Transformers. Transformers utilize parallel attention heads and hierarchical filters to perform computation and inference Vaswani et al. (2017). Consequently, attempting to identify critical parameters at a fine-grained level is often inaccurate, as this approach fails to capture the broader contextual relationships inherent in Transformers.

In this paper, we propose a **Structure-aware Parameter-Efficient Unlearning** (SPE-Unlearn) approach that targets influence-critical parameters at the structural level for Transformers. Specifically, SPE-Unlearn formulates the unlearning objective through a pair of learnable masks applied to heads and filters. The derivation for this formulation ensures the effective influence removal and guides the identification of key structures. These masks are further refined by considering intra-layer interactions, and a warm-start greedy search algorithm is employed to efficiently optimize the process. Equipped with these structure-aware masks, we integrate SPE-Unlearn into second-order unlearning updates. While second-order unlearning introduces an approximation error, sparse updates using structure-aware masks can mitigate the errors, thereby preserving overall model performance. In addition, we analyze that structure-aware masks can demonstrate significant advantages in successive settings (Hu et al. (2023); Liu et al. (2023a)). In this context, we are the first to categorize second-order successive setting into two types based on whether intermediate information from previous removal requests is retained: **memory-free unlearning** (Guo et al. (2020); Gu et al. (2024)) and **memory-aided unlearning** (Liu et al. (2023a)). Our approach demonstrates exceptional robustness by effectively containing errors within selected structures, especially in memory-free unlearning scenarios. Our key contributions are summarized as follows:

- We introduce a new paradigm for identifying influence-critical parameters in Transformers, SPE-Unlearn, which operates at the structural level. Our approach theoretically derives importance scores for selecting key structures using a pair of learnable masks. These structure-aware masks can be seamlessly integrated into existing unlearning methods.

- We integrate SPE-Unlearn into second-order unlearning and analyze the gains with structure-aware masks. Extensive experiments across diverse datasets using three models demonstrate proposed method offers a superior trade-off among efficacy, fidelity, and efficiency.

- We categorize successive unlearning into two successive scenarios: memory-free unlearning and memory-aided unlearning. Empirical studies show that unlearning with structure-aware masks can handle a greater number of removal requests compared to standard unlearning before retraining becomes necessary, especially in memory-free scenarios.

## 2 PRELIMINARY

### 2.1 PROBLEM FORMULATIONS

Machine unlearning aims to remove the influence of targeted data from a trained model. Let $\mathcal{D} = \{x_i\}_{i=1}^M$ denote a training dataset containing $M$ data points, where each $x_i$ corresponds to an individual data point. Starting with the original model $\theta^*$ which was trained on $\mathcal{D}$, the objective of unlearning is to effectively remove the sensitive or compliance-related data while maintaining overall performance. Specifically, for the unlearning task, the dataset $\mathcal{D}$ is grouped into two subsets: **forgetting dataset** $\mathcal{D}_f$ and **remaining dataset** $\mathcal{D}_r$, i.e., $\mathcal{D} = \mathcal{D}_f \cup \mathcal{D}_r$ . The forgetting dataset $\mathcal{D}_f$ consists of the targeted data we aim to remove from the model. Accordingly, the remaining dataset $\mathcal{D}_r$ includes the data we intend to retain and potentially further optimize. Given a loss function $\ell$ for targeted task, the objective of unlearning can be framed as learning an optimal model $\theta_U^*$:

$$\theta_U^* = \arg\min_\theta \mathcal{L}(\theta; \mathcal{D}_r) = \arg\min_\theta \sum_{x \in \mathcal{D}_r} \ell(\theta; x) + \lambda\Omega(\theta), \tag{1}$$

where $\mathcal{L}(\theta; \mathcal{D}_r)$ represents the total loss on the dataset $\mathcal{D}_r$ with $\theta$, and $\lambda\Omega(\theta)$ is a common regularization term (Hart et al. (2000)). The most viable solution to address this optimization problem is retraining the model from scratch. However, retraining can be costly in terms of time and computing resources. A practical alternative, known as the second-order unlearning update (Guo et al. (2020); Golatkar et al. (2020); Izzo et al. (2021); Warnecke et al. (2021); Liu et al. (2024b)), deduces the general close-form parameter modification from the original model $\theta^*$:

$$\theta \approx \theta^* + \mathbf{H}_{\theta^*}^{-1} \sum_{x \in \mathcal{D}_f} \nabla_\theta \ell(\theta^*; x), \tag{2}$$

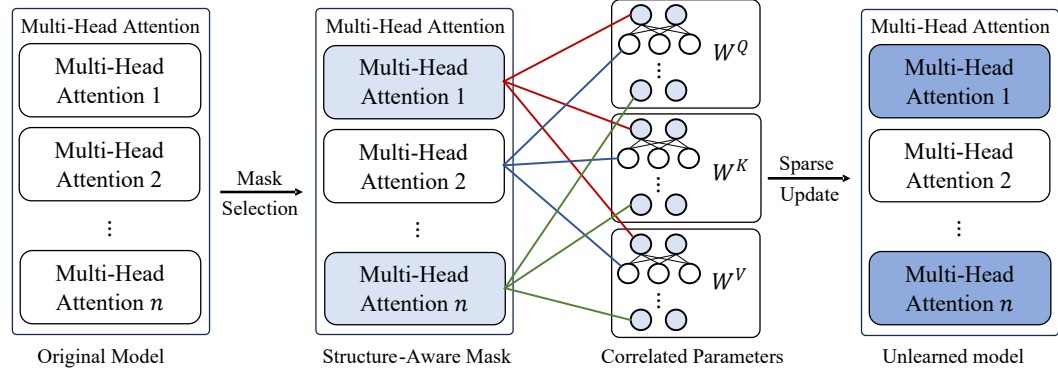

Figure 1: Illustration of our method applied to obtain important heads. Starting with the original model, key heads are identified highlighted in light blue. The different colored dashed lines (e.g., red, blue, green) represent the connections between heads and their correlated parameters. Last, we update the active parameters within heads highlighted in blue to represent unlearning process.

where $\mathbf{H}_{\theta^*}^{-1}$ is the inverse of the Hessian matrix $\nabla_\theta^2 \mathcal{L}(\theta^*; \mathcal{D}_r)$ evaluated at $\theta^*$. This method is derived from influence function (Koh & Liang (2017)), which provides a bounded approximation error to facilitate effective unlearning (Guo et al. (2020)).

However, second-order unlearning involves the inverse Hessian computation, which is highly sensitive to parameters. Given the large number of parameters in large-scale models, this unlearning method cannot be applied directly. A common practice to approximate it is using the empirical FIM (Peste et al. (2021); Liu et al. (2024a); Gu et al. (2024)). Additionally, studies (Amari et al. (2019)) have shown that the off-diagonal elements of the FIM tend to be much smaller than the diagonal elements, usually by a factor $\frac{1}{\sqrt{n}}$, where $n$ represents the dimension of the FIM. This insight highlights the effectiveness of using a diagonal approximation, particularly in large models with vast parameter counts (Hwang (2024)). As a result, we further adopt the empirical diagonal FIM $\widehat{\mathcal{I}}$ to approximate the Hessian matrix:

$$\widehat{\mathcal{I}}(\theta; \mathcal{D}) = \frac{1}{|\mathcal{D}|} \sum_{x \in \mathcal{D}} \nabla \ell(\theta; x)^2. \tag{3}$$

The storage of the diagonal FIM requires only $\mathcal{O}(d)$ space, and the inverse operation takes only $\mathcal{O}(d)$ time, where $d$ denotes the number of model parameters. This makes second-order unlearning method straightforward and efficient to implement.

## 3 STRUCTURE-AWARE PARAMETER-EFFICIENT MACHINE UNLEARNING

Inspired by the lottery hypothesis (Frankle & Carbin (2018)), recent research suggests that localizing functional regions within neural networks can make the model more effective for specific tasks (Zhang et al. (2024b)). Given the high dimension for large models, empirically identifying influence-critical parameters from a too fine-grained perspective is both inefficient and often sub-optimal. To this end, we propose SPE-Unlearn, which derived a pair of masks to pinpoint influence-critical parameters within heads and filters in Section 3.1. By selectively targeting the most influence-critical parameters, SPE-Unlearn is integrated into second-order unlearning in 3.2, enabling more efficient and effective unlearning processes. At last, we extend SPE-Unlearn to support successive unlearning, demonstrating its robustness in Section 3.3.

### 3.1 STRUCTURE-AWARE PARAMETER LOCALIZATION

While parameter-efficient methods involve identifying critical parameters, this process can be framed as finding an optimal binary mask. In this context, a mask value of 1 indicates that the corresponding parameter should be updated, while a value of 0 represents that the corresponding parameter should remain frozen. Given that the number of structures is significantly smaller than the number of parameters (e.g., 37K vs. 110M in case of BERT-base), SPE-Unlearn adapt a coarse-grained method to pinpoint influence-critical parameters in heads and filters. Thus, we formulate

the unlearning objective (1) with a learnable pair of masks for the heads and filters as a constrained optimization problem. To streamline the problem, we provide a general expression for the heads and filters by introducing the mask variables m:

$$\mathrm{m}^* = \arg\min_{\mathrm{m}} \mathcal{L}(\mathrm{m}; \theta^*, \mathcal{D}_\mathrm{r}) \quad \text{s.t.} \frac{\sum_{i=1}^{|\mathrm{m}|} \mathrm{m}_i}{|\mathrm{m}|} < 1 - \mathrm{S}, \tag{4}$$

where $|\mathrm{m}|$ is the number of mask variables, $\theta^*$ represents the original model, and S denotes the sparsity (e.g., 90%) which determines the proportion of frozen structures. Since we focus exclusively on the mask variables, we henceforth regard the parameters $\theta^*$ as constants. Thus, the total loss $\mathcal{L}(\theta; \mathcal{D}_\mathrm{r})$ can be mapped to $\mathcal{L}(\mathrm{m}; \theta^*, \mathcal{D}_\mathrm{r})$. If $\mathcal{L}$ is differentiable with respect to m, we then approximate $\mathcal{L}(\mathrm{m}; \theta^*, \mathcal{D}_\mathrm{r})$ using the second-order Taylor series around the mask variables $\mathbb{1}$:

$$\mathcal{L}(\mathrm{m}; \theta^*, \mathcal{D}_\mathrm{r}) \approx \mathcal{L}(\mathbb{1}; \theta^*, \mathcal{D}_\mathrm{r}) - (\mathbb{1} - \mathrm{m})\nabla_\mathrm{m}\mathcal{L}(\mathbb{1}; \theta^*, \mathcal{D}_\mathrm{r}) + \frac{1}{2}(\mathbb{1} - \mathrm{m})^\mathrm{T}\nabla_\mathrm{m}^2\mathcal{L}(\mathbb{1}; \theta^*, \mathcal{D}_\mathrm{r})(\mathbb{1} - \mathrm{m}). \tag{5}$$

As the original model $\theta^*$ has converged to a local minimum of $\nabla_\mathrm{m}\mathcal{L}(\mathbb{1}; \theta^*, \mathcal{D})$, we can assume that $\nabla_\mathrm{m}\mathcal{L}(\mathbb{1}; \theta^*, \mathcal{D}) = 0$ (LeCun et al. (1989)). Incorporating this assumption, we simplify gradient term in the Taylor series approximation, i.e., $\nabla_\mathrm{m}\mathcal{L}(\mathbb{1}; \theta^*, \mathcal{D}_\mathrm{r}) = \nabla_\mathrm{m}\mathcal{L}(\mathbb{1}; \theta^*, \mathcal{D}) - \sum_{x \in \mathcal{D}_\mathrm{f}} \nabla_\mathrm{m}\ell(\mathbb{1}; \theta^*, x) = -\sum_{x \in \mathcal{D}_\mathrm{f}} \nabla_\mathrm{m}\ell(\mathbb{1}; \theta^*, x)$. As $\mathcal{L}(\mathbb{1}; \theta^*, \mathcal{D}_\mathrm{r})$ is a constant, we can adjust the unlearning objective with mask variables:

$$\mathrm{m}^* \approx \arg\min_{\mathrm{m}} (\mathbb{1} - \mathrm{m}) \sum_{x \in \mathcal{D}_\mathrm{f}} \nabla_\mathrm{m}\ell(\mathbb{1}; \theta^*, x) + \frac{1}{2}(\mathbb{1} - \mathrm{m})^\mathrm{T}\nabla_\mathrm{m}^2\mathcal{L}(\mathbb{1}; \theta^*, \mathcal{D}_\mathrm{r})(\mathbb{1} - \mathrm{m}). \tag{6}$$

Thus, the optimization problem depends on the two factors: the gradient with respect to the forgetting dataset $\mathcal{D}_\mathrm{f}$ (i.e., $\sum_{x \in \mathcal{D}_\mathrm{f}} \nabla_\mathrm{m}\ell(\mathbb{1}; \theta^*, x)$) and the Hessian matrix with respect to the remaining dataset $\mathcal{D}_\mathrm{r}$ (i.e., $\nabla_\mathrm{m}^2\mathcal{L}(\mathbb{1}; \theta^*, \mathcal{D}_\mathrm{r})$). These components together reflect the effectiveness of influence removal. Since forming the Hessian matrix directly is computationally prohibitive, we approximate it using the empirical **diagonal** FIM of the mask variables with Equation (3). This leads to a simplified form of the optimization objective in Equation (6):

$$\mathrm{m}^* \approx \arg\min_{\mathrm{m}} (\mathbb{1} - \mathrm{m}) \sum_{x \in \mathcal{D}_\mathrm{f}} \nabla_\mathrm{m}\ell(\mathbb{1}; \theta^*, x) + \frac{1}{2}(\mathbb{1} - \mathrm{m})^2 \widehat{\mathcal{I}}(\mathbb{1}; \theta^*, \mathcal{D}_\mathrm{r}). \tag{7}$$

Given that the mask variable can only be set to 0 or 1, we transform the optimization problem into a mask selection problem with heads and filters:

$$\mathrm{m}^* \approx \arg\min_{\mathrm{m}} \sum_i \left[ (1 - \mathrm{m}_i)\big[ \sum_{x \in \mathcal{D}_\mathrm{f}} \nabla_\mathrm{m}\ell(\mathbb{1}; \theta^*, x)\big]_i + \frac{1}{2}(1 - \mathrm{m}_i)^2 \big[\widehat{\mathcal{I}}(\mathbb{1}; \theta^*, \mathcal{D}_\mathrm{r})\big]_i \right]. \tag{8}$$

Therefore, we propose importance scores to identify influence-critical heads and filters. Each head or filter can be assessed based on the sum of its corresponding gradient and half of the diagonal FIM element. Heads or filters with higher scores will be prioritized for selection. Additionally, to better understand the influence of off-diagonal elements on mask selection for each layer, we replace the diagonal FIM with the block diagonal FIM, where each block is associated with a layer. Thus, Equation (7) decomposes into *layer-wise* optimization problems:

$$\mathrm{m}_l^* \approx \arg\min_{\mathrm{m}_l} (\mathbb{1} - \mathrm{m}_l)\big[ \sum_{x \in \mathcal{D}_\mathrm{f}} \nabla_\mathrm{m}\ell(\mathbb{1}; \theta^*, x)\big]_l + \frac{1}{2}(\mathbb{1} - \mathrm{m}_l)^2 \big[\widehat{\mathcal{I}}(\mathbb{1}; \theta^*, \mathcal{D}_\mathrm{r})\big]_l, \tag{9}$$

where $l$ represents the layer being optimized. This optimization problem can be efficiently solved using a greedy search with warm start (Kwon et al. (2022)), i.e., initializing the mask variables $\mathrm{m}_l$ derived from Equation (8). In this process, we iteratively swap unselected each head (or filter) with the highest importance score for selected one in the current mask to further optimize Equation (9), yielding an approximate solution after one round of swapping. Consequently, the rearranged mask variables captures the impact of intra-layer interactions, enabling precise localization of the parameters within the model structures. Additionally, our approach can be integrated with other methods for identifying influence-critical parameters, offering enhanced flexibility. Detailed information about these techniques can be found in Appendix A.4. In practice, our derivation can also be applied to other unlearning objectives, such as maximizing the loss on the forgetting dataset (Jia et al. (2024)). Detailed information is presented in Appendix B.

---

**Algorithm 1** Structure-aware Parameter-Efficient Second-Order Unlearning

---

**Input:** remaining dataset $\mathcal{D}_r$, forgetting dataset $\mathcal{D}_f$, Transformer model $T$, loss function $\ell$, model parameter $\theta$, sparsity S, unlearning rate $\eta$
**Output:** Updated model parameter $\theta$

1: Initialize mask $m \leftarrow \mathbb{1}$, parameter FIM $\widehat{\mathcal{I}} \leftarrow 0$, parameter gradients $g_\theta \leftarrow 0$, mask gradients $g_m^r \leftarrow [], g_m^f \leftarrow []$
2: **for** each $x$ in $\mathcal{D}_r$ **do**                                     ▷ Iterate data points in $\mathcal{D}_r$
3:     $\nabla_m \ell(\theta, x), \nabla_\theta \ell(\theta, x) \leftarrow T(\theta, m, \ell, x)$
4:     $\widehat{\mathcal{I}} += \frac{1}{|\mathcal{D}_r|} \nabla_\theta \ell(\theta, x)^2$          ▷ Obtain the parameter diagonal FIM in $\mathcal{D}_r$
5:     Append $\nabla_m \ell(\theta, x)$ to $g_m^r$            ▷ Gather the mask gradients in $\mathcal{D}_r$
6: **end for**
7: **for** each $x$ in $\mathcal{D}_f$ **do**                                   ▷ Iterate data points in $\mathcal{D}_f$
8:     $\nabla_m \ell(\theta, x_i), \nabla_\theta \ell(\theta, x_i) \leftarrow T(\theta, m, \ell, x_i)$
9:     $g_\theta += \nabla_\theta \ell(\theta, x_i)$           ▷ Obtain the parameter gradient in $\mathcal{D}_f$
10:     Append $\nabla_m \ell(\theta, x_i)$ to $g_m^f$        ▷ Gather the mask gradients in $\mathcal{D}_f$
11: **end for**
12: $SC \leftarrow \frac{1}{2}(g_m^r)^2 + g_m^f$         ▷ Compute importance scores of structures
13: $IN \leftarrow$ indices of unimportant heads         ▷ Find the optimal mask indices
14: $IN^* \leftarrow$ rearrange the mask indices with warm start
15: $m[IN^*] = 0$                       ▷ Set unimportant indices to 0
16: $\theta += \eta * m \circ \widehat{\mathcal{I}}^{-1} g_\theta$       ▷ Sparse Second-Order unlearning update
17: **return** $\theta$

---

## 3.2 STRUCTURE-AWARE PARAMETER-EFFICIENT SECOND-ORDER UNLEARNING

By pinpointing influence-critical parameters within the heads and filters, `SPE-Unlearn` enables efficient integration with widely-adopted unlearning methods, e.g., fine-tuning (Golatkar et al. (2020)) and gradient difference (Liu et al. (2022); Jia et al. (2024)). A key observation is that both `SPE-Unlearn` and second-order unlearning share the computational need for gradients and FIM. Therefore, we leverage second-order unlearning as a representative case study to showcase the efficacy of our approach.

Following the insights of `SPE-Unlearn`, we formalize **Structure-aware Parameter-Efficient Second-Order unlearning** (SPE-SO) by introducing sparse mask variables linked to the outputs of heads and filters:

$$\theta \approx \theta^* + m \circ \left[ [\widehat{\mathcal{I}}(\theta^*; \mathcal{D}_r)]^{-1} \sum_{x \in \mathcal{D}_f} \nabla_\theta \ell(\theta^*; x) \right], \tag{10}$$

where $m$ are the binary mask variables, and $\circ$ denotes the Hadamard product. Note that Equation 2 can be represented by setting all mask variables to 1. This method introduces several key advantages over standard unlearning techniques. First, by incorporating sparsity through structure-aware masks, SPE-SO significantly reduces the number of parameters required for the expensive computation of the Hessian matrix. This leads to lower computational complexity, making the method more scalable and efficient when applied to large-scale models. Second, SPE-SO offers a more tightly bounded approximation error compared to standard methods. The approximation error is reduced by a factor that is directly proportional to the sparsity introduced by the mask variables. This ensures that the unlearning process remains highly accurate while avoiding unnecessary parameter updates. Furthermore, by restricting the influence-critical parameters within the heads and filters, SPE-SO provides fine-grained control over the error bounds.

Algorithm 1 presents the workflow of SPE-SO, which handles removal requests by accumulating and processing them collectively. The algorithm can be adapted to various constraints, such as time or memory. For scenarios where computational efficiency is the primary concern, SPE-SO allows for pre-computation of the gradient and diagonal FIM for the entire training dataset. Upon receiving removal requests, we can compute the data information about forgetting dataset to obtain the required data, i.e., the gradient of forgetting dataset and diagonal FIM of remaining dataset. Alternatively, to reduce memory consumption, SPE-SO can retrieve only the necessary information by utilizing selected structures tied to specific parameters.

### 3.3 STRUCTURE-AWARE PARAMETER-EFFICIENT SUCCESSIVE UNLEARNING

Successive unlearning presents a practical scenario where data owners request the removal of data points from the model at intervals, necessitating prompt deletion (e.g., machine learning as a service (MLaaS) (Hu et al. (2023))). While prior work has proposed different approaches to successive unlearning, we introduce the classification to better differentiate how unlearning algorithm is used. Specifically, we categorize second-order successive unlearning into two distinct types based on whether or not the algorithm retains information from removed data: memory-free (Guo et al. (2020); Gu et al. (2024)) and memory-aided (Liu et al. (2023a)).

Memory-free unlearning iteratively update the **latest model** following each removal request without retaining any information from the removed data. However, this method increases the unlearning approximation error, as the updates are based solely on the latest model, which can be more severe for Transformers. In contrast, memory-aided unlearning retains data information (i.e., gradients and FIM) to efficient unlearn on the **original model**. In what follows, we apply structure-aware masks into these successive unlearning scenarios and discuss the advantages of these masks.

#### 3.3.1 MEMORY-FREE UNLEARNING

The way to apply `SPE-Unlearn` into the memory-free unlearning is straightforward. Upon each data removal request, we can directly compute the structure-aware mask and apply second-order unlearning. Specifically, the model is progressively updated based on the state from the previous unlearning cycle. At timestamp $t$ (i.e., the $t$-th unlearning request), structure-aware parameter-efficient memory-free unlearning can be formalized:

$$\mathrm{m}^t \circ \left[ [\widehat{\mathcal{I}}(\theta^{t-1}; \mathcal{D}_{\mathrm{r}}^t)]^{-1} \nabla_\theta \ell(\theta^{t-1}; x^t) \right], \tag{11}$$

where $\theta^{t-1}$ represents the unlearned model parameters at timestamp $t-1$, $\mathcal{D}_{\mathrm{r}}^t$ and $x^t$ denote the remaining dataset and the data point to be removed at timestamp $t$. Additionally, $\mathrm{m}^t$ is the structure-aware mask corresponding to the $t$-th removal request.

Although memory-free unlearning is simple and easy to implement, it suffers a major drawbacks. This method inherently diverges from the Taylor series approximation, which tends to introduce small errors during each approximation. As these errors accumulate with each successive update, the model is continually adjusted based on its latest state rather than retaining the original form. Consequently, with an increasing number of removal requests, the disparity between the original and updated models widens, resulting in a gradual decline in model performance.

Once the number of unlearning requests surpasses a certain threshold, the model needs to be retrained from scratch to recover its performance (detailed in Table 1). Fortunately, structure-aware masks allows for more removal requests before retraining becomes necessary (as shown in Figure 4). This improvement is likely due to selectively adjust only the structures directly related to the data being removed. By confining the cumulative errors to a minimal subset of parameters, the overall impact on the model performance is reduced. Consequently, the model remains robust even after multiple unlearning operations, delaying the need for costly retraining.

Table 1: Accuracy results using memory-free unlearning with standard second-order unlearning under varying removal requests.

| Removal Requests | 1 | 4 | 8 | 10 |
|---|---|---|---|---|
| Testing Accuracy | 84.34% | 83.86% | 83.6% | 83.46% |
| Remaining Accuracy | 94.33% | 94.18% | 94.05% | 93.86% |

#### 3.3.2 MEMORY-AIDED UNLEARNING

Compared to memory-free unlearning, memory-aided unlearning approximates directly through a Taylor expansion at original model parameters. In contrast, memory-aided unlearning (Liu et al. (2023a)) accumulates the gradients on forgotten data and FIM on remaining dataset to achieve unlearning. Specifically, upon receiving the $t$-th unlearning request, structure-aware parameter-efficient memory-aided unlearning at timestamp $t$ can be expressed as follows:

$$\mathrm{m}^t \circ \left\{ \left[ \frac{|\mathcal{D}_{\mathrm{r}}^{t-1}| \cdot \widehat{\mathcal{I}}(\theta^*; \mathcal{D}_{\mathrm{r}}^{t-1}) - \widehat{\mathcal{I}}(\theta^*; x^t)}{|\mathcal{D}_{\mathrm{r}}^{t-1} - 1|} \right]^{-1} \left[ \sum_{x \in \mathcal{D}_{\mathrm{f}}^{t-1}} \nabla_\theta \ell(\theta^*; x) + \nabla_\theta \ell(\theta^*; x^t) \right] \right\}, \tag{12}$$

where $\mathcal{D}_\mathrm{f}^{t-1}$ represents the data points that have already been removed at timestamp $t-1$, $\mathcal{D}_\mathrm{r}^{t-1}$ denotes the remaining dataset at timestamp $t-1$. In practice, rather than storing these data points directly, we retain the gradients or FIM associated with the data in memory. With each new unlearning request, these data information are updated accordingly. Furthermore, considering the proportion of the forgetting dataset is negligible, the mask selection process can be accelerated. As a result, in the mask selection Equation (8), the term $\sum_{x\in\mathcal{D}_\mathrm{f}} \nabla_m \ell(\mathbb{1}; x)$ can be omitted, and the term $\widehat{\mathcal{I}}(\mathbb{1}; \mathcal{D}_\mathrm{r})$ can be approximated by $\widehat{\mathcal{I}}(\mathbb{1}; \mathcal{D})$, resulting in the following simplification:

$$\mathrm{m}^* \approx \arg\min_{\mathrm{m}} \sum_i (\mathbb{1} - \mathrm{m}_i)^2 \widehat{\mathcal{I}}(\mathbb{1}; \mathcal{D})_i, \tag{13}$$

Since the Equation (13) is derived based solely on the entire dataset, the corresponding mask can be per-computed during the pre-unlearning phase. Although this simplification enhances efficiency, it does not fully account for the influence of the data points slated for deletion. Thus, we finally rearrange the mask variables using Equation (9), which allows for a more targeted adjustment. In memory-aided scenario, unlearning is achieved through a single-step second-order update on the original model. Therefore, the key strength of structure-aware masks stems from the superiority of `SPE-Unlearn` in handling general second-order unlearning, which offers a tighter approximation error bound to facilitate more effective and accurate data removal.

# 4 EXPERIMENTS

## 4.1 EXPERIMENT SETUPS

**Models and Datasets.** We conduct comprehensive experiments on three pretrained Transformer models: BERT-base (Devlin et al. (2018)), DistilBERT (Sanh et al. (2019)), and RoBERTa-large (Liu et al. (2019)). These models are accessed through the HuggingFace Transformers library (Wolf et al. (2020)). Our evaluation spans four GLUE tasks (MNLI, QQP, SST-2, and STS-B) (Wang et al. (2018)) and two SQuAD tasks (SQuAD v1.1 and SQuAD v2.0) (Rajpurkar (2016)). Consistent with the configurations outlined in prior works Devlin et al. (2018); Sanh et al. (2019); Liu et al. (2019), we fine-tune these models, treating them as the original models for our experiments.

**Unlearning methods.** Our experiments focus on comparing the proposed method SPE-SO with several other unlearning methods. These methods include Fine-Tuning (FT), Gradient Difference (GD) (Liu et al. (2022), Jia et al. (2024)), Sparsity-Aware unlearning (SA) (Liu et al. (2024a)). For FT, we continue training the original model on the remaining dataset for 3 epochs. For GD, the model is fine-tuned on entire dataset for 3 epochs, with the gradient direction reversed for the data that needs to be forgotten. For SA, fine-tuning is performed on the remaining dataset with a sparsity penalty ($\gamma = 5e - 5$) applied to the parameters for 3 epochs. Additionally, we also include the standard Second-Order unlearning (SO) method, which excludes structure-aware masks, to evaluate the effectiveness of `SPE-Unlearn`. Meanwhile, Retraining from scratch (RT) serves as the gold standard, where the model is fine-tuned on the remaining dataset following the configurations from Devlin et al. (2018); Sanh et al. (2019); Liu et al. (2019). Detailed hyperparameters are presented in Appendix A.1.

**Evaluation metrics.** We analyze the unlearning methods from three aspects (Warnecke et al. (2021); Gu et al. (2024)): 1) **Efficacy** in removing the targeted data. We evaluate this using unlearning accuracy and membership inference attacks (MIA) on $\mathcal{D}_\mathrm{f}$. Unlearning accuracy directly reflects the effectiveness of the unlearning algorithm, while MIA assesses the vulnerability of the model to attacks after unlearning. In practice, we use a confidence-based MIA predictor to gauge the likelihood of a successful attack (Liu et al. (2024a); Song et al. (2019)); 2) **Fidelity** of model utility. We measure this by examining both the remaining accuracy and the testing accuracy to assess the preservation of model performance and its generalization ability after unlearning; 3) **Efficiency** of executing the unlearning methods. We report the time required to perform unlearning as a measure of speed and computational efficiency.

## 4.2 EXPERIMENT RESULTS

We present the experimental results using the MNLI dataset as a case study. Detailed results for additional datasets are provided in Appendix A. Unless otherwise specified, the default number of

Table 2: Overall results of unlearning performance using different unlearning methods under three fine-tuned models. We focus on 90% sparsity SPE-SO as our approaches.

| Model | Method | Efficacy | | Fidelity | | Efficiency |
|---|---|---|---|---|---|---|
| | | Unlearning Accuracy ↓ | MIA ↓ | Remaining Accuracy ↑ | Testing Accuracy ↑ | Time ↓ |
| BERT-base | RT | 85.16% | 0.7500 | 97.95% | 84.78% | 8880s |
| | FT | 92.19% | 0.8594 | **99.16%** | **84.63%** | 5651s |
| | GD | 90.62% | 0.8437 | 99.13% | 84.20% | 5690s |
| | SA | 89.84% | 0.8437 | 92.77% | 82.05% | 4845s |
| | SO | **85.94%** | 0.8047 | 94.07% | 84.60% | 1160s |
| | SPE-SO | **85.94%** | **0.7969** | 94.15% | 84.62% | 1274s |
| DistilBERT | RT | 82.81% | 0.7266 | 96.61% | 82.47% | 4989s |
| | FT | 94.53% | 0.8906 | **98.94%** | **81.63%** | 2434s |
| | GD | 91.41% | 0.8750 | 98.72% | 81.37% | 2498s |
| | SA | 90.62% | 0.8750 | 96.49% | 81.23% | 2399s |
| | SO | 89.06% | 0.8516 | 96.37% | 81.29% | 587s |
| | SPE-SO | **88.28%** | **0.8359** | 96.47% | 81.62% | 643s |
| RoBERTa-large | RT | 90.62% | 0.8125 | 98.79% | 90.02% | 62068s |
| | FT | 97.66% | 0.9766 | 99.50% | **90.02%** | 18004s |
| | GD | 95.31% | **0.8906** | **99.64%** | 89.57% | 18176s |
| | SA | 92.97% | **0.8906** | 96.86% | 87.08% | 14634s |
| | SO | 92.97% | **0.8906** | 94.32% | 88.99% | 3575s |
| | SPE-SO | **92.19%** | **0.8906** | 95.75% | 89.52% | 3642s |

unlearned samples is 128. We randomly select 128 samples as the forgetting dataset $\mathcal{D}_f$ and use all orthogonal samples as the remaining dataset $\mathcal{D}_r$. In what follows, we compare different unlearning methods and conduct an in-depth analysis of our approach.

**Structure-Aware sparse unlearning is effective.** Table 2 presents the unlearning performance of various unlearning methods across three models. As subsequent experiments show that 90% sparsity is sufficient for effective unlearning, we focus on the SPE-SO with 90% sparsity regime for comparison with other methods. Our experiments reveal that FT is inefficient for unlearning in Transformers, while SA demonstrates strong

Table 3: Memory consumption with three models. SPE-SO takes 90% sparsity.

| Memory (MB) | BERT-base | DistilBERT | RoBERTa-large |
|---|---|---|---|
| SO | 995.7 | 544.0 | 3371.6 |
| SPE-SO | 663.8 | 377.4 | 2174.9 |

unlearning efficacy but at the cost of significantly compromising model fidelity. GD generally strikes a reasonable balance between efficacy and efficiency. However, these methods demands considerable time due to the lengthy fine-tuning process. In contrast, both SO and SPE-SO achieve effective unlearning with just a single epoch over the dataset, which provide robust efficacy guarantees with minimal impact on fidelity. As shown in Table 3, we further compare memory usage during model updates for SO and SPE-SO. Although SPE-SO takes more time to identify structure-aware mask, it has lower storage overhead and delivers superior performance compared to SO. Thus, we conclude that SPE-SO with 90% sparsity is sufficient to strike a favorable "efficacy-fidelity-efficiency" trade-off.

**A sparsity of 90% is sufficient for effective unlearning.** We explore the effectiveness of various sparsity strategies in facilitating unlearning. Figure 2 shows the relationship between testing accuracy and sparsity while maintaining comparable unlearning efficacy. As sparsity increases up to 90%, the model retains high utility. However, when sparsity surpasses 90%, a sharp decline in model accuracy occurs, indicating that updating fewer than 10% of parameters may be insufficient to preserve utility. Similar effects of sparsity strategies on unlearning performance are observed across other datasets (detailed in Appendix A.3). We also delve into the functional regions responsible for unlearning within models, but find no single network layer that stands out as particularly crucial for unlearning. This suggests that the effectiveness of unlearn-

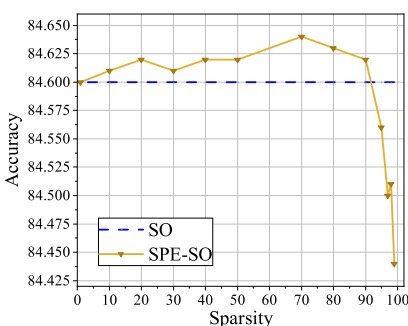

Figure 2: Testing accuracy of SO and SPE-SO applied to BERT-base across varying sparsity.

ing may be task-specific, resisting any fixed structural or parametric approach. Overall, our findings

emphasize that a 90% sparsity strategy strikes the sufficient balance between efficiency and effectiveness in unlearning tasks, offering a practical approach without compromising much utility.

**Selective parameter updates only in filters can effectively accomplish unlearning.** (Pochinkov & Schoots (2024)) argued that pruning filters is more effective than heads. To further investigate this claim, we conducted a comparative analysis of three selective parameter update strategies: heads-only, filters-only, and heads&filters in Figure 3. All the experiments are designed to provide comparable unlearning guarantees varying sparsity. While the heads-only approach demonstrated superior testing accuracy at moderate sparsity levels (30% to 70%), it falls behind in terms of remaining accuracy. In contrast, the filters-only strategy not only maintained stability at lower sparsity but also delivered consistently strong unlearning performance at higher sparsity. Notably, we observed that compared to updating the parameter both in heads and filters, updating only the parameters in either heads or filters can achieve better unlearning performance. This underscores that more focused updates may mitigate unnecessary overhead, without sacrificing performance. Among the approaches, filters-only updates consistently proved to be the most stable and effective, making it a more optimal choice for robust unlearning.

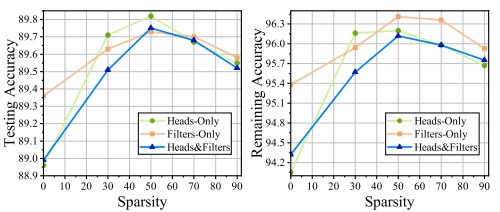
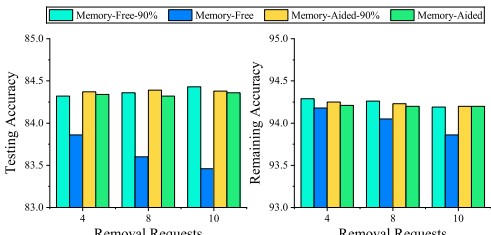

Figure 3: Testing accuracy and remaining accuracy for various sparsity applied to RoBERTa-large after unlearning different structures.

Figure 4: Results using memory-free unlearning and memory-aided unlearning on BERT-base under varying removal requests.

**Structure-aware masks benefit robust unlearning.** Our method highlights that structure-aware masks serve as an effective mechanism for guiding data removal, enabling models to meet strict unlearning guarantees while preserving model performance. Motivated by this observation, we further explore the potential of structure-aware masks in successive unlearning scenarios, focusing on both memory-free and memory-aided unlearning, as depicted in Figure 4. Our results show that sparse updates with structure-aware offer marginal improvements over full updates in memory-aided unlearning. This is likely because memory-aided unlearning operates by updating the model directly from its original state in a single step, minimizing the relative advantage of sparse updates. In contrast, sparse updates offer significant benefits in memory-free unlearning. When all parameters are updated in memory-free unlearning, model fidelity is overly impacted, consistent with the analysis in Section 3.3.1. However, applying sparse updates with 90% sparsity in memory-free unlearning preserves high model utility, even after 10 removal requests. This suggests that structure-aware masks can support a higher volume of removal requests before retraining becomes necessary. These results highlight the potential of structure-aware masks to enhance the robustness of unlearning.

## 5 RELATED WORK

**Transformer Unlearning.** The concept of machine unlearning was first introduced by (Cao & Yang (2015)). Initially applied to simple model, machine unlearning has since been extended to Transformer models (Jang et al. (2022); Eldan & Russinovich (2023); Yao et al. (2023; 2024); Chen et al. (2024); Jia et al. (2024); Gu et al. (2024)). (Jang et al. (2022)) proposed inverting the training objective on forgetting sequences and utilize straightforward gradient ascent. As gradient ascent significantly degrades performance, (Yao et al. (2024)) refined the objective function by employing gradient descent on in-distribution data to enhance robustness. Subsequently, (Jia et al. (2024)) provided a comprehensive overview of unlearning objectives and developed a second-order optimization unlearning approach. (Gu et al. (2024)) further investigated the effectiveness of second-order updates in Transformers. However, these methods primarily focus on updating all model parameters, which is computationally expensive and time-consuming. In our work, we study the parameter-efficient methods to achieve effective unlearning in Transformers.

**Parameter-efficient Unlearning.** Parameter-efficient unlearning methods focus on identifying influence-critical parameters and updating only those to accelerate the unlearning process. Several strategies (Ma et al. (2022); Pochinkov & Schoots (2024); Shi et al. (2023); Liu et al. (2023b); Wu & Harandi (2024); Foster et al. (2024); Schoepf et al. (2024)) have been proposed to assess parameter importance. Although these approaches may be applicable to Transformers, they are largely heuristic or empirical, which can result in sub-optimal outcomes for unlearning tasks. Recently, (Liu et al. (2024a)) highlighted that unlearning can be effective when performed on a pruned model with a theoretical foundation. However, pruning primarily focuses on identifying parameters critical to maintain model performance, which does not align with the desiderata of unlearning. Additionally, the focus on parameter ignore the complex intra-structural interactions within Transformers, which results in inaccurate identification of the parameters. Therefore, we specifically target at heads and filters within Transformers and derive an efficient strategy to identify influence-critical parameters.

## 6    CONCLUDING REMARKS

In this work, we propose structure-aware parameter-efficient unlearning (`SPE-Unlearn`), a novel approach tailored for Transformers. `SPE-Unlearn` derives an optimal masking strategy to identify influence-critical parameters within heads and filters. By selectively targeting these key parameters, `SPE-Unlearn` infuses into second-order unlearning update to demonstrate its efficacy and strengths. We further analyze the advantages of our method across both memory-free and memory-aided successive unlearning scenarios. Empirical study demonstrate that our method accommodates more removal requests than standard second-order unlearning in memory-free unlearning scenarios. Additionally, comprehensive experiments conducted on various Transformer models and datasets reveal that our method with 90% sparsity outperforms existing approaches.

For future work, we suggest extending to other existing unlearning methods to demonstrate the effectiveness of `SPE-Unlearn` in Transformers. While our experiments focus on small-scale Transformers, we plan to explore larger-scale models (e.g, OPT-13b and LLaMA2-13b) to better understand the behavior of structure-aware masks. Furthermore, our study concentrates on fine-tuned models, which limits the ability to unlearn deeply ingrained undesired information from pre-trained models. To address this, we aim to identify structure-aware masks directly in pre-trained models.

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

## A    ADDITIONAL EXPERIMENTAL DETAILS

### A.1    HYPERPARAMETERS

We fine-tune BERT-base, DistilBERT, and RoBERTa-large on different datasets using AdamW with weight decay of 0 as vanilla models. The learning rate is selected from $10^{-5}, 2 \cdot 10^{-5}, 3 \cdot 10^{-5}$ and $5 \cdot 10^{-5}$. Such learning rate is also applied to unlearning methods such as Retraining, Fine-Tuning (FT), Structure-Aware unlearning (SA), and Gradient Difference (GD). The number of epochs is set to 5. For unlearning, the number of epochs is fixed at 3 for FT, SA, and GD, while it is fixed at 5 for retraining. The unlearning rate for Second-Order unlearning (SO) are chosen through grid search in the range $[10^{-6}, 10^{-7}]$. For Structure-aware Parameter-Efficient Second-Order SO (SPE-SO), the unlearning rate increases proportionally with the fraction of updated parameters relative to the total parameters compared to SO.

### A.2    COMPARE TO OTHER UNLEARNING METHODS

We compare ours to other unlearning methods in three GLUE tasks (QQP, SST-2 and STS-B) and two SQuAD taks (SQuAD v1.1 and SQuAD v2.0) under three models (detailed in Table 4 to Table 8). The evaluation metrics vary depending on the task. For example, we use Spearman correlations to assess STS-B, while F1 scores are reported for both SQuAD v1.1 and SQuAD v2.0. Higher values for these metrics indicate better model performance.

### A.3    FIND THE APPROPRIATE SPARSITY

We aim to determine the level of sparsity that can ensure adequate model performance while providing sufficient unlearning guarantees. Therefore, we conducted a detailed sparsity analysis on additional datasets, as shown in Figure 5. Our results indicate that updating all parameters is not the most effective strategy for unlearning, as it can lead to excessive forgetting, causing a rapid decline in model performance. In contrast, we found that a sparsity of 50% offers the most efficient improvement in unlearning. Moreover, sparsity levels between 80% and 90% perform on par with, and sometimes even surpass, the performance of other methods.

### A.4    IDENTIFY INFLUENCE-CRITICAL PARAMETERS IN STRUCTURES

In our approach, the mask is applied to specific heads and filters, resulting in a relatively coarse granularity for unlearning. To achieve a more refined and precise method, we further investigate the importance of individual parameters within these selected heads and filters. Our hypothesis is that by focusing on individual parameters, we can identify more fine-grained regions that are critical for effective unlearning.

Table 4: Overall results of unlearning performance using different unlearning methods under three fine-tuned models on QQP dataset.

| Model | Method | Efficacy | | Fidelity | | Efficiency |
|---|---|---|---|---|---|---|
| | | Unlearning Accuracy ↓ | MIA ↓ | Remaining Accuracy ↑ | Testing Accuracy ↑ | Time ↓ |
| BERT-base | RT | 92.97% | 0.875 | 98.48% | 91.38% | 9560s |
| | FT | 96.09% | 0.9219 | 99.56% | 91.26% | 5759s |
| | GD | 96.09% | 0.9219 | **99.58%** | **91.29%** | 5858s |
| | SA | 92.19% | **0.8906** | 92.52% | 88.52% | 5579s |
| | SO | 92.97% | 0.9063 | 97.69% | 90.65% | 832s |
| | SPE-SO | **92.19%** | **0.8906** | 98.03% | 90.72% | 926s |
| DistilBERT | RT | 90.62% | 0.8203 | 98.52% | 90.39% | 6291s |
| | FT | 98.44% | 0.9453 | 99.61% | **90.25%** | 3619s |
| | GD | 96.09% | 0.9297 | **99.65%** | 90.16% | 3763s |
| | SA | 95.31% | 0.9141 | 96.80% | 81.14% | 3571s |
| | SO | 92.19% | 0.8594 | 98.23% | 90.05% | 415s |
| | SPE-SO | **91.41%** | **0.8594** | 98.36% | 90.12% | 468s |
| RoBERTa-large | RT | 91.41% | 0.8594 | 99.17% | 92.19% | 79214s |
| | FT | 98.44% | 0.9609 | 99.85% | **92.18%** | 21742s |
| | GD | 94.53% | 0.9453 | **99.91%** | 92.14% | 23239s |
| | SA | 93.75% | **0.8750** | 98.69% | 91.48% | 20793s |
| | SO | **92.97%** | **0.8750** | 98.93% | 91.56% | 2598s |
| | SPE-SO | **92.87%** | **0.8750** | 98.86% | 91.46% | 2639s |

Table 5: Overall results of unlearning performance using different unlearning methods under three fine-tuned models on SST-2 dataset.

| Model | Method | Efficacy | | Fidelity | | Efficiency |
|---|---|---|---|---|---|---|
| | | Unlearning Accuracy ↓ | MIA ↓ | Remaining Accuracy ↑ | Testing Accuracy ↑ | Time ↓ |
| BERT-base | RT | 93.75% | 0.9297 | 99.06% | 93.00% | 915s |
| | FT | 96.88% | 0.9609 | 99.25% | 92.78% | 479s |
| | GD | 95.31% | **0.8984** | **99.53%** | 92.78% | 518s |
| | SA | 95.31% | 0.9062 | 98.82% | 89.79% | 450s |
| | SO | 94.53% | 0.9141 | 98.96% | 92.89% | 93s |
| | SPE-SO | **94.53%** | **0.8984** | 98.93% | **93.35%** | 103s |
| DistilBERT | RT | 92.97% | 0.8984 | 98.78% | 91.40% | 403s |
| | FT | 95.31% | **0.8594** | 97.67% | 90.37% | 238s |
| | GD | **94.53%** | 0.8984 | **98.90%** | 90.25% | 243s |
| | SA | **94.53%** | 0.9141 | 98.28% | 89.91% | 235s |
| | SO | **94.53%** | 0.8906 | 96.35% | **91.63%** | 46s |
| | SPE-SO | **94.53%** | **0.8906** | 96.35% | **91.63%** | 53s |
| RoBERTa-large | RT | 94.53% | 0.9063 | 99.64% | 96.10% | 4698s |
| | FT | 97.66% | 0.9753 | **99.44%** | **96.22%** | 1430s |
| | GD | 93.75% | 0.9219 | 97.98% | 95.33% | 1492s |
| | SA | 94.53% | 0.9219 | 98.84% | 95.07% | 1423s |
| | SO | **94.53%** | 0.9297 | 99.14% | 94.15% | 311s |
| | SPE-SO | **94.53%** | **0.8984** | 99.45% | 94.55% | 320s |

To implement this, we adopt Wanda (Sun et al. (2023)) as our selection mechanism. Wanda operates by analyzing the forgetting dataset, which serves as the input for the selective process. The values returned by Wanda represent the importance of each neuron to the unlearning task—higher values indicate neurons that are more critical for unlearning. After this analysis, we apply a sparsity of 90% on SPE-SO, selecting the most important parameters to retain based on their Wanda scores. These selected parameters are then targeted for unlearning. This method not only aligns with the broader goal of structural selection but also enhances the precision of unlearning by targeting specific neurons within the model. Detailed results of this selective mechanism are shown in Table 9.

However, Our experimental results indicate that incorporating the parameter selection mechanism does not improve unlearning performance in SPE-SO. We hypothesize that this outcome stems from the inherent complexity of balancing unlearning precision with model utility. While selecting individual parameters based on their Wanda scores allows for a more targeted and theoretically precise unlearning process, this fine-grained approach may inadvertently reduce the overall model's adaptability and robustness.

Table 6: Overall results of unlearning performance using different unlearning methods under three fine-tuned models on STS-B dataset.

| Model | Method | Efficacy | | Fidelity | | Efficiency |
|---|---|---|---|---|---|---|
| | | Unlearning Spearman Corr. ↓ | MIA ↓ | Remaining Spearman Corr. ↑ | Testing Spearman Corr. ↑ | Time ↓ |
| BERT-base | RT | 86.60% | 0.5156 | 97.86% | 88.63% | 148s |
| | FT | 95.37% | 0.8750 | 96.72% | 88.49% | 76s |
| | GD | 91.66% | 0.594 | 99.17% | 88.50% | 84s |
| | SA | 98.70% | 0.8750 | **99.31%** | **88.60%** | 64s |
| | SO | 86.91% | 0.632 | 98.00% | 87.76% | 9s |
| | SPE-SO | **86.47%** | **0.5234** | 98.24% | 87.76% | 10s |
| DistilBERT | RT | 87.31% | 0.6563 | 93.10% | 85.45% | 62s |
| | FT | 91.15% | **0.6875** | **95.20%** | 85.16% | 29s |
| | GD | 89.12% | 0.7031 | 94.81% | **85.37%** | 30s |
| | SA | 92.36% | 0.7109 | 93.85% | 85.26% | 27s |
| | SO | **87.61%** | **0.6875** | 91.71% | 85.02% | 5s |
| | SPE-SO | 87.75% | 0.703125 | 92.01% | 85.26% | 5.5s |
| RoBERTa-large | RT | 90.97% | 0.5781 | 97.95% | 92.01% | 671s |
| | FT | 96.19% | 0.7656 | **98.68%** | **91.92%** | 198s |
| | GD | 92.18% | 0.5703 | 96.17% | 90.33% | 205s |
| | SA | 96.25% | 0.7344 | **98.68%** | 91.57% | 176s |
| | SO | 91.28% | 0.5078 | 97.46% | 91.57% | 31s |
| | SPE-SO | **91.13%** | **0.484375** | 97.88% | 91.79% | 35s |

Table 7: Overall results of unlearning performance using different unlearning methods under three fine-tuned models on SQuAD v1.1 dataset.

| Model | Method | Efficacy | | Fidelity | | Efficiency |
|---|---|---|---|---|---|---|
| | | Unlearning F1 ↓ | MIA ↓ | Remaining F1 ↑ | Testing F1 ↑ | Time ↓ |
| BERT-base | RT | 87.62% | 0.5938 | 95.23% | 88.18% | 6328s |
| | FT | 92.36% | 0.7578 | 96.38% | 87.73% | 3765s |
| | GD | 87.27% | 0.6797 | **96.44%** | 87.34% | 3775s |
| | SA | 89.75% | 0.7031 | 91.94% | 86.85% | 3800s |
| | SO | **86.26%** | **0.5625** | 94.33% | **87.74%** | 764s |
| | SPE-SO | 86.74% | 0.5781 | 94.25% | 87.60% | 809s |
| DistilBERT | RT | 84.38% | 0.5391 | 94.34% | 85.35% | 3203s |
| | FT | 92.54% | 0.7188 | 97.49% | 85.09% | 1936s |
| | GD | 87.18% | 0.6016 | **97.54%** | 85.05% | 1956s |
| | SA | 89.52% | 0.7109 | 96.42% | 83.86% | 1921s |
| | SO | 85.79% | **0.5547** | 93.51% | 85.35% | 763s |
| | SPE-SO | **85.35%** | **0.5547** | 93.65% | **85.37%** | 812s |
| RoBERTa-large | RT | 90.41% | 0.6484 | 97.92% | 92.50% | 18439s |
| | FT | 94.74% | 0.7734 | 98.97% | **93.15%** | 11365s |
| | GD | 91.75% | 0.6484 | **99.15%** | 92.98% | 11520s |
| | SA | 91.05% | 0.6875 | 95.16% | 89.36% | 11116s |
| | SO | **90.71%** | **0.500** | 94.93% | 90.95% | 2008s |
| | SPE-SO | 90.81% | 0.5234 | 95.14% | 91.03% | 2141s |

# B  IDENTIFY KEY STRUCTURES IN OTHER UNLEARNING OBJECTIVE

Machine unlearning typically relies on the specific unlearning objective to design optimization algorithms. For instance, second-order unlearning is achieved by minimizing the loss on the remaining dataset (i.e., Equation 2). To simplify the optimization, a Taylor expansion of the unlearning objective is performed on the original model. Following the optimization process, we identify the influence-critical parameters using a structure-aware approach in Transformers, which then guides the second-order unlearning update.

The mainstream class of existing Large Language Model (LLM) unlearning methods also follow the pattern of optimization based on the objective function. Gradient Ascent (GA) (Jang et al. (2022)) aims to maximize the loss for the forgetting dataset. Building on this, Gradient Difference (GD) (Liu et al. (2022)) further strives to maintain performance on the remaining dataset. Direct Preference Optimization (DPO) (Rafailov et al. (2024)) seeks to algin the model by replacing the original response on forgetting dataset with the alternative answers "I don't know". Inspired by DPO, Negative

Table 8: Overall results of unlearning performance using different unlearning methods under three fine-tuned models on SQuAD v2.0 dataset.

| Model | Method | Efficacy | | Fidelity | | Efficiency |
| --- | --- | --- | --- | --- | --- | --- |
| | | Unlearning F1 ↓ | MIA ↓ | Remaining F1 ↑ | Testing F1 ↑ | Time ↓ |
| BERT-base | RT | 73.77% | 0.6484 | 98.72% | 75.77% | 9560s |
| | FT | 88.80% | 0.8047 | **98.84%** | **74.52%** | 5532s |
| | GD | 81.54% | 0.7344 | 90.28% | 74.22% | 5600s |
| | SA | 79.16% | 0.6797 | 96.03% | 72.65% | 5512s |
| | SO | 78.03% | 0.6797 | 93.66% | 73.33% | 1043s |
| | SPE-SO | **77.40%** | **0.6563** | 93.90% | 73.57% | 1123s |
| DistilBERT | RT | 71.86% | 0.6641 | 93.75% | 69.80% | 4715s |
| | FT | 89.78% | 0.8047 | **97.55%** | **69.71%** | 2880s |
| | GD | 79.93% | 0.7188 | 97.28% | 68.16% | 2894s |
| | SA | 80.89% | 0.7497 | 95.76% | 68.46% | 2863s |
| | SO | 77.73% | 0.7109 | 92.10% | 68.36% | 415s |
| | SPE-SO | **76.30%** | **0.7031** | 91.82% | 67.95% | 468s |
| RoBERTa-large | RT | 87.03% | 0.7734 | 98.42% | 86.58% | 27053s |
| | FT | 89.15% | 0.7891 | **98.01%** | **85.89%** | 16466s |
| | GD | 88.26% | 0.7734 | 97.93% | 85.37% | 16652s |
| | SA | **84.05%** | 0.7343 | 93.21% | 80.82% | 13470s |
| | SO | 87.70% | **0.7188** | 94.68% | 85.22% | 3092s |
| | SPE-SO | 87.34% | **0.7188** | 94.76% | 85.50% | 3390s |

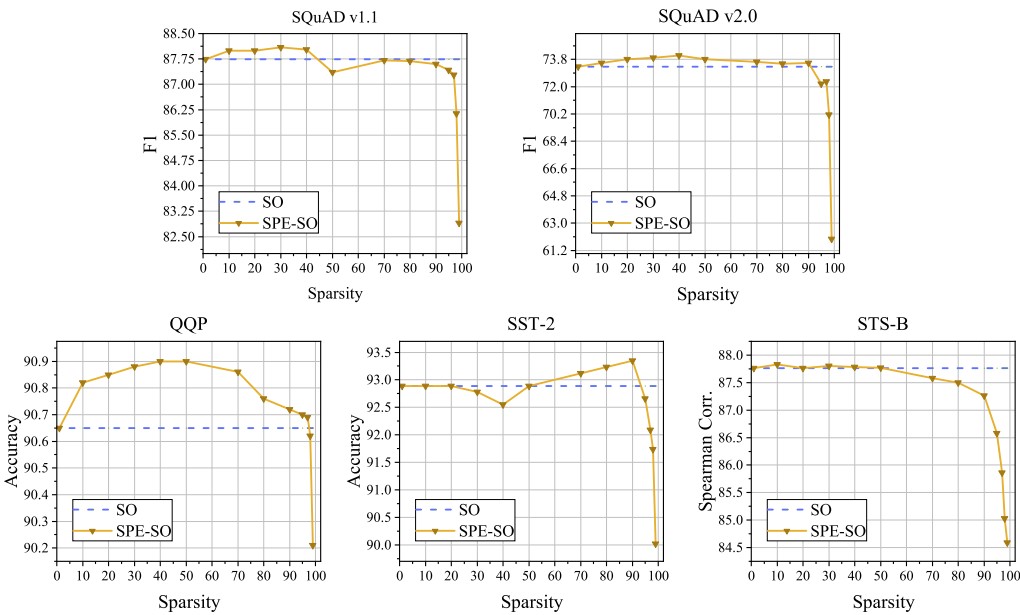

Figure 5: Accuracy of SO and SPE-SO applied to BERT-base across varying sparsity on additional datasets.

Preference Optimization (NPO) (Zhang et al. (2024a)) targets maximizing the discrepancy on the forgetting dataset between the original model and unlearned model. Although the objective functions of these methods differ, their optimization approach relies on gradient ascent, which aims to maximize the loss on the forgetting dataset. We express the unlearning objective in the following form:

$$\arg\max_{\theta} \mathcal{L}(\theta; \mathcal{D}_{\mathrm{f}}), \tag{14}$$

Table 9: Overall results of unlearning performance are presented using SPE-SO on BERT-base, both with and without the neuron selection mechanism. For clarity, SPE-SO denotes SPE-SO applied to structures at 90% sparsity, while SPE-SO(90%) indicates SPE-SO applied to both structures and parameters, each with 90% sparsity.

| Datasets | Method | Efficacy | | Fidelity | | Efficiency |
|---|---|---|---|---|---|---|
| | | Unlearning Accuracy ↓ | MIA ↓ | Remaining Accuracy ↑ | Testing Accuracy ↑ | Time ↓ |
| MNLI | SPE-SO | 85.94% | 0.7969 | **94.15%** | **84.62%** | 1274s |
| | SPE-SO(90%) | 85.94% | 0.7969 | 94.12% | 84.61% | 1280s |
| QQP | SPE-SO | 92.19% | 0.9062 | **98.03%** | **90.72%** | 926s |
| | SU(90%) | 92.19% | **0.8828** | 97.67% | 90.46% | 930s |
| SST-2 | SPE-SO | 94.53% | **0.8984** | **98.93%** | 93.35% | 103s |
| | SPE-SO(90%) | 94.53% | 0.9141 | 98.92% | 93.35% | 105s |
| STS-B | SPE-SO | **86.47%** | **0.632** | **97.24%** | **87.26%** | 10s |
| | SPE-SO(90%) | 86.87% | 0.6406 | 96.36% | 86.99% | 11s |
| SQuAD v1.1 | SPE-SO | **85.74%** | **0.5781** | **94.25%** | **87.60%** | 809s |
| | SPE-SO(90%) | 86.16% | 0.5859 | 93.98% | 87.19% | 812s |
| SQuAD v2.0 | SPE-SO | 77.40% | 0.6563 | **93.90%** | 73.57% | 1123s |
| | SPE-SO(90%) | 77.40% | 0.6563 | 93.75% | **73.73%** | 1128s |

where $\mathcal{D}_f$ is the forgetting dataset. We observe that this objective is similar to minimizing the loss on the remaining dataset and can also identify the influence-critical parameters using a comparable approach. First, we introduce a learnable pair of masks for heads and filters:

$$\mathbf{m}^* = \arg\max_{\mathbf{m}} \mathcal{L}(\mathbf{m}; \theta^*, \mathcal{D}_f) \quad \text{s.t.} \frac{\sum_{i=1}^{|\mathbf{m}|} \mathbf{m}_i}{|\mathbf{m}|} < 1 - S, \tag{15}$$

where $|\mathbf{m}|$ is the number of mask variables, $\theta^*$ represents the original model, and S denotes the sparsity. We then approximate it using the second-order Taylor series around the mask variables $\mathbb{1}$:

$$\mathcal{L}(\mathbf{m}; \theta^*, \mathcal{D}_f) \approx \mathcal{L}(\mathbb{1}; \theta^*, \mathcal{D}_f) - (\mathbb{1} - \mathbf{m})\nabla_{\mathbf{m}}\mathcal{L}(\mathbb{1}; \theta^*, \mathcal{D}_f) + \frac{1}{2}(\mathbb{1} - \mathbf{m})^{\mathrm{T}}\nabla_{\mathbf{m}}^2\mathcal{L}(\mathbb{1}; \theta^*, \mathcal{D}_f)(\mathbb{1} - \mathbf{m}). \tag{16}$$

We then use the diagonal FIM to approximate the Hessian matrix and omit constant terms, resulting in a simplified optimization objective:

$$\mathbf{m}^* \approx \arg\max_{\mathbf{m}}(\mathbb{1} - \mathbf{m}) \sum_{x \in \mathcal{D}_r} \nabla_{\mathbf{m}}\ell(\mathbb{1}; \theta^*, x) + \frac{1}{2}(\mathbb{1} - \mathbf{m})^2 \widehat{\mathcal{I}}(\mathbb{1}; \theta^*, \mathcal{D}_f). \tag{17}$$

Since the mask can only take values of 0 or 1, we can derive the importance evaluation function:

$$\mathbf{m}^* \approx \arg\max_{\mathbf{m}} \sum_i \left[ (1 - \mathbf{m}_i)\big[ \sum_{x \in \mathcal{D}_r} \nabla_{\mathbf{m}}\ell(\mathbb{1}; \theta^*, x) \big]_i + \frac{1}{2}(1 - \mathbf{m}_i)^2 \big[ \widehat{\mathcal{I}}(\mathbb{1}; \theta^*, \mathcal{D}_f) \big]_i \right]. \tag{18}$$

After obtaining the initial mask, we further optimize the objective using the block diagonal FIM to rearrange mask:

$$\mathbf{m}_l^* \approx \arg\max_{\mathbf{m}_l}(\mathbb{1} - \mathbf{m}_l)\big[ \sum_{x \in \mathcal{D}_r} \nabla_{\mathbf{m}}\ell(\mathbb{1}; \theta^*, x) \big]_l + \frac{1}{2}(\mathbb{1} - \mathbf{m}_l)^2 \big[ \widehat{\mathcal{I}}(\mathbb{1}; \theta^*, \mathcal{D}_f) \big]_l. \tag{19}$$

where $l$ represents the layer being optimized. Equipped with the identified key structures, we facilitate four LLM unlearning methods.

## B.1 EXPERIMENTS

We evaluate unlearning methods on the Task of Fictitious Unlearning (TOFU) dataset (Maini et al. (2024)) using LLama2-7b-chat model (Touvron et al. (2023)). The unlearning scenarios of TOFU can be divided into three types: Forget01, Forget05, and Forget10, which represent forgetting dataset proportions of 1%, 5%, and 10% of the total dataset, respectively. The baseline includes seven methods: Retraining (RT), Fine-tuning (FT), Sparsity-Aware Unlearning (SA), GA, GD, DPO and NPO. We apply three structure-aware parameter-efficient unlearning methods into GA, GD, DPO

and NPO for comparison. These methods include: 1) maximize the loss on the forgotten dataset (MLF) as the unlearning objective, 2) minimize the loss of the remaining dataset (MLR) as the forgetting objective (i.e. the original method in Section 3.1), and 3) use the norm of the gradients associated with the structure to evaluate its importance (NORM).

**Experimental details.** We use AdamW with a weight decay of 0.01 and a learning rate of $10^{-5}$ in RT, FT, SA, GA, GD, DPO and NPO. Besides, we set the learning rate for structure-aware parameter-efficient methods to $2 \cdot 10^{-4}$ or $3 \cdot 10^{-4}$. In addition, the sparsity of structure-aware parameter-efficient methods is 90%. All the experiments run for 5 epochs. We also use three main aspects (i.e., efficacy, fidelity and efficiency) to evaluate the unlearning performance. We use Rouge scores, normalized probabilities, and the True Ratios on the forgotten dataset to measure efficacy, and those metrics on the real authors, world facts, and remaining dataset to measure fidelity. We still use unlearning time to evaluate efficiency. Note that smaller values do not necessarily indicate better forgetting performance. The goal for unlearning is to closely match that achieved through retraining.

**Results.** We find that performing FT and SA only on the remaining dataset does not meet the unlearning requirements. Although the original GA, GD, and DPO methods can achieve unlearning, they all exhibit severe catastrophic forgetting on the Forget10 dataset. In contrast, NPO is the most efficient among these methods. Furthermore, our experiments indicate that sparse updates are better suited for unlearning than full updates, as they offer a stronger guarantee of unlearning while more effectively preserving performance, even on the Forget10 dataset. Additionally, the NORM-based method significantly reduces computation time, but it is less effective than the MLF-based and MLR-based methods. We observe that the MLR-based method offers a robust balanced trade-off among unlearning efficacy, model fidelity, and computational efficiency.

Table 10: Overall results of unlearning performance using different unlearning methods under LLama2-7b-chat on TOFU Forget01. 'Prob.' indicates the normalized probabilities, 'TR' represents the True Ratios. Forget quality (FQ) and Model Utility (MU) are also used to evaluate the efficacy and fidelity respectively.

| Method | Efficacy | | | | Fidelity | | | | | | | | | | Efficiency |
| | Forgetting Dataset | | | FQ↑ | Real Authors | | | World Facts | | | Remaining Dataset | | | MU↑ | Time↓ |
| | Rouge | Prob. | TR | | Rouge↑ | Prob.↑ | TR↑ | Rouge↑ | Prob.↑ | TR↑ | Rouge↑ | Prob.↑ | TR↑ | | |
| RT | 0.39 | 0.18 | 0.69 | 1.0 | 0.93 | 0.45 | 0.58 | 0.88 | 0.41 | 0.54 | 0.99 | 0.99 | 0.47 | 0.62 | - |
| FT | 0.96 | 0.99 | 0.53 | 5.04e-4 | 0.94 | 0.45 | 0.58 | 0.87 | 0.42 | 0.55 | 0.97 | 0.99 | 0.48 | 0.62 | 95.19s |
| SA | 0.95 | 0.99 | 0.53 | 1.88e-4 | 0.93 | 0.45 | 0.58 | 0.87 | 0.42 | 0.56 | 0.98 | 0.99 | 0.48 | 0.62 | 94.88s |
| GA | 0.49 | **0.23** | 0.54 | 1.27e-3 | 0.92 | 0.42 | 0.55 | **0.89** | 0.41 | 0.54 | 0.92 | 0.95 | **0.49** | 0.60 | 96.13s |
| MLF-GA | 0.64 | 0.83 | 0.54 | 1.27e-3 | **0.93** | **0.45** | **0.58** | 0.88 | 0.43 | **0.57** | **0.97** | **0.98** | 0.48 | **0.63** | 149.46s |
| MLR-GA | **0.43** | 0.55 | **0.56** | 1.27e-3 | **0.93** | **0.45** | **0.58** | 0.89 | 0.44 | **0.57** | 0.93 | 0.96 | 0.48 | **0.63** | 147.53s |
| NORM-GA | 0.57 | 0.79 | 0.54 | 1.27e-3 | **0.93** | **0.45** | **0.58** | 0.88 | 0.43 | **0.57** | 0.96 | **0.98** | 0.48 | **0.63** | **77.07s** |
| GD | 0.55 | **0.53** | 0.53 | 1.27e-3 | 0.94 | 0.44 | 0.57 | 0.86 | 0.42 | 0.55 | **0.96** | **0.98** | 0.48 | 0.61 | 220.57s |
| MLF-GD | 0.64 | 0.83 | 0.53 | 1.27e-3 | 0.94 | **0.45** | 0.59 | 0.88 | **0.43** | 0.56 | **0.96** | **0.98** | 0.48 | **0.63** | 174.66s |
| MLR-GD | **0.48** | 0.61 | **0.54** | 1.27e-3 | 0.94 | **0.45** | 0.57 | 0.88 | **0.43** | 0.56 | 0.94 | **0.98** | 0.48 | 0.62 | 172.73s |
| NORM-GD | 0.64 | 0.83 | 0.53 | 1.27e-3 | 0.94 | **0.45** | 0.59 | 0.88 | **0.43** | 0.56 | 0.96 | **0.98** | 0.48 | **0.63** | **102.12s** |
| DPO | 0.69 | 0.92 | **0.58** | 5.04e-4 | 0.93 | **0.48** | **0.62** | 0.88 | **0.45** | 0.56 | 0.94 | 0.98 | 0.46 | **0.64** | 380.96s |
| MLF-DPO | 0.69 | 0.83 | 0.54 | 5.04e-4 | 0.94 | 0.45 | 0.58 | 0.88 | 0.43 | 0.56 | **0.96** | 0.98 | **0.48** | 0.63 | 169.90s |
| MLR-DPO | **0.65** | **0.81** | 0.54 | 1.88e-4 | 0.94 | 0.45 | 0.58 | 0.88 | 0.43 | 0.56 | **0.96** | 0.98 | 0.48 | 0.63 | 237.73s |
| NORM-DPO | 0.69 | 0.83 | 0.54 | 5.04e-4 | **0.94** | 0.45 | 0.58 | 0.87 | 0.43 | 0.56 | **0.96** | 0.98 | **0.48** | 0.63 | **167.12s** |
| NPO | **0.52** | **0.27** | **0.55** | 3.02e-3 | 0.92 | 0.42 | 0.55 | 0.87 | 0.41 | 0.54 | 0.94 | 0.95 | 0.49 | 0.61 | 253.88s |
| MLF-NPO | 0.59 | 0.68 | 0.54 | 1.27e-3 | **0.94** | **0.45** | **0.58** | 0.89 | 0.44 | **0.57** | **0.96** | **0.98** | 0.48 | **0.63** | 174.66s |
| MLR-NPO | 0.55 | 0.75 | 0.54 | 1.27e-3 | 0.93 | **0.45** | **0.58** | 0.89 | 0.44 | 0.56 | 0.95 | **0.98** | 0.48 | **0.63** | 196.31s |
| NORM-NPO | 0.59 | 0.78 | 0.54 | 1.27e-3 | 0.93 | **0.45** | **0.58** | 0.89 | **0.44** | 0.56 | 0.95 | **0.98** | 0.48 | **0.63** | **125.70s** |

Table 11: Overall results of unlearning performance using different unlearning methods under LLama2-7b-chat on TOFU Forget05. 'Prob.' indicates the normalized probabilities, 'TR' represents the True Ratios. Forget quality (FQ) and Model Utility (MU) are also used to evaluate the efficacy and fidelity respectively.

| Method | Efficacy | | | | Fidelity | | | | | | | | | | Efficiency |
| | Forgetting Dataset | | | FQ↑ | Real Authors | | | World Facts | | | Remaining Dataset | | | MU↑ | Time↓ |
| | Rouge | Prob. | TR | | Rouge↑ | Prob.↑ | TR↑ | Rouge↑ | Prob.↑ | TR↑ | Rouge↑ | Prob.↑ | TR↑ | | |
| RT | 0.39 | 0.15 | 0.67 | 1.0 | 0.96 | 0.42 | 0.55 | 0.90 | 0.40 | 0.53 | 0.98 | 0.99 | 0.46 | 0.62 | - |
| FT | 0.92 | 0.97 | 0.51 | 8.33e-16 | 0.94 | 0.47 | 0.61 | 0.89 | 0.44 | 0.57 | 0.93 | 0.96 | 0.48 | 0.63 | 404.03s |
| SA | 0.97 | 0.99 | 0.51 | 3.43e-16 | 0.94 | 0.45 | 0.58 | 0.87 | 0.42 | 0.55 | 0.98 | 0.99 | 0.48 | 0.62 | 404.03s |
| GA | 0.10 | 3.62e-3 | **0.65** | **4.31e-4** | 0.63 | 0.35 | 0.49 | 0.85 | 0.40 | 0.53 | 0.17 | 0.02 | 0.46 | 0.11 | 404.27s |
| MLF-GA | **0.20** | **3.93e-3** | 0.62 | 1.18e-4 | **0.86** | **0.41** | **0.56** | **0.88** | 0.41 | 0.57 | **0.40** | **0.31** | 0.46 | **0.48** | 346.03s |
| MLR-GA | 0.17 | 2.31e-3 | 0.61 | 4.75e-5 | 0.83 | 0.42 | 0.58 | 0.86 | **0.42** | **0.58** | 0.33 | 0.17 | 0.45 | 0.42 | 342.01s |
| NORM-GA | 0.18 | 1.08e-3 | 0.56 | 1.21e-10 | 0.85 | 0.40 | 0.55 | 0.86 | 0.40 | 0.57 | 0.36 | 0.22 | 0.44 | 0.44 | **252.45s** |
| GD | 0.30 | 1.79e-2 | 0.54 | 2.83e-4 | 0.79 | 0.35 | 0.49 | **0.87** | 0.38 | 0.53 | 0.46 | 0.42 | **0.50** | 0.49 | 1009.22s |
| MLF-GD | **0.37** | **0.15** | 0.61 | 2.83e-4 | **0.94** | 0.44 | 0.59 | 0.86 | **0.43** | 0.57 | 0.81 | **0.91** | 0.48 | **0.61** | 463.18s |
| MLR-GD | 0.33 | 2.31e-2 | **0.64** | **0.63** | 0.90 | **0.46** | **0.60** | 0.87 | **0.43** | 0.57 | 0.79 | 0.88 | 0.48 | **0.61** | 460.68s |
| NORM-GD | **0.37** | **0.15** | 0.61 | 2.83e-4 | **0.94** | 0.44 | 0.59 | 0.86 | **0.43** | 0.57 | 0.81 | **0.91** | 0.48 | **0.61** | **371.12s** |
| DPO | 4.57e-2 | 0.64 | **0.62** | 6.57e-12 | 0.57 | 0.46 | 0.60 | 0.83 | **0.46** | 0.57 | 0.23 | 0.73 | 0.40 | 0.47 | 1800.04s |
| MLF-DPO | **0.30** | 0.19 | 0.60 | 8.06e-7 | **0.92** | 0.46 | 0.59 | 0.86 | 0.45 | **0.58** | 0.76 | **0.88** | 0.48 | **0.61** | 782.51s |
| MLR-DPO | 0.28 | 9.35e-2 | **0.62** | **1.84e-4** | 0.90 | 0.46 | 0.60 | **0.87** | 0.45 | **0.58** | 0.69 | **0.88** | 0.48 | **0.61** | 780.65s |
| NORM-DPO | 0.19 | **0.14** | 0.61 | 4.75e-5 | 0.88 | 0.46 | 0.60 | 0.82 | 0.45 | **0.58** | 0.66 | 0.78 | **0.48** | 0.60 | **691.08s** |
| NPO | 0.34 | 0.11 | **0.66** | **1.18e-4** | **0.94** | 0.33 | 0.42 | **0.89** | 0.38 | 0.49 | 0.42 | 0.36 | 0.46 | 0.46 | 1183.55s |
| MLF-NPO | 0.33 | 0.12 | 0.61 | 1.11e-5 | 0.91 | **0.44** | **0.58** | 0.88 | **0.42** | **0.57** | 0.74 | 0.83 | **0.48** | **0.60** | 568.75s |
| MLR-NPO | **0.34** | **0.15** | 0.59 | 8.11e-8 | 0.90 | **0.44** | **0.58** | 0.87 | **0.42** | **0.57** | 0.76 | **0.85** | 0.48 | **0.60** | 570.66s |
| NORM-NPO | 0.32 | 0.12 | 0.61 | 1.11e-5 | 0.91 | 0.43 | **0.58** | 0.88 | **0.42** | 0.56 | 0.74 | 0.82 | **0.48** | 0.59 | **481.09s** |

Table 12: Overall results of unlearning performance using different unlearning methods under LLama2-7b-chat on TOFU Forget10. 'Prob.' indicates the normalized probabilities, 'TR' represents the True Ratios. Forget quality (FQ) and Model Utility (MU) are also used to evaluate the efficacy and fidelity respectively.

| Method | Efficacy | | | | Fidelity | | | | | | | | | | Efficiency |
| | Forgetting Dataset | | | FQ↑ | Real Authors | | | World Facts | | | Remaining Dataset | | | MU↑ | Time↓ |
| | Rouge | Prob. | TR | | Rouge↑ | Prob.↑ | TR↑ | Rouge↑ | Prob.↑ | TR↑ | Rouge↑ | Prob.↑ | TR↑ | | |
| RT | 0.41 | 0.15 | 0.67 | 1.0 | 0.92 | 0.43 | 0.57 | 0.90 | 0.41 | 0.54 | 0.98 | 0.99 | 0.47 | 0.61 | - |
| FT | 0.89 | 0.96 | 0.51 | 2.43e-19 | 0.94 | 0.48 | 0.62 | 0.89 | 0.45 | 0.58 | 0.89 | 0.96 | 0.47 | 0.64 | 827.26s |
| SA | 0.98 | 0.99 | 0.50 | 1.69e-15 | 0.92 | 0.44 | 0.58 | 0.86 | 0.41 | 0.55 | 0.98 | 0.99 | 0.49 | 0.62 | 832.32s |
| GA | 1.19e-3 | 6.26e-33 | 0.79 | 5.40e-18 | 0.0 | 0.25 | 0.21 | 0 | 0.25 | 0.20 | 0.01 | 1.57e-32 | 0.12 | 0.0 | 822.49s |
| MLF-GA | 0.14 | 3.25e-4 | 0.56 | 2.06e-13 | 0.72 | 0.47 | **0.67** | 0.73 | 0.46 | **0.61** | 0.21 | 2.51e-2 | 0.40 | 0.16 | 590.08s |
| MLR-GA | **0.22** | **2.79e-2** | **0.62** | **0.34** | **0.84** | **0.49** | 0.66 | **0.87** | **0.46** | 0.59 | **0.36** | **0.33** | **0.46** | **0.51** | 588.77s |
| NORM-GA | 0.15 | 3.02e-4 | 0.54 | 1.45e-14 | 0.68 | 0.47 | **0.67** | 0.73 | 0.46 | 0.60 | 0.20 | 3.31e-2 | 0.41 | 0.19 | **464.81s** |
| GD | 1.31e-2 | 3.01e-18 | **0.70** | 1.07e-13 | 0.49 | 0.46 | 0.63 | 0.82 | 0.44 | 0.58 | 0.25 | 0.24 | 0.48 | 0.42 | 2042.64s |
| MLF-GD | **0.31** | 1.85e-2 | 0.59 | **7.31e-3** | **0.89** | 0.48 | 0.61 | **0.87** | 0.46 | 0.59 | **0.61** | 0.75 | 0.47 | 0.60 | 847.88s |
| MLR-GD | **0.31** | **1.85e-2** | 0.59 | **7.31e-3** | **0.89** | 0.48 | 0.61 | **0.87** | 0.46 | 0.59 | **0.61** | 0.75 | **0.49** | 0.60 | 843.89s |
| NORM-GD | 0.30 | 1.53e-2 | 0.59 | 3.11e-3 | **0.89** | **0.51** | **0.67** | 0.85 | 0.46 | **0.61** | 0.53 | 0.69 | 0.48 | 0.60 | **719.93s** |
| DPO | 1.05e-2 | 0.51 | **0.66** | 1.49e-9 | 5.33e-3 | 0.43 | 0.57 | 0.17 | 0.43 | 0.53 | 1.17e-2 | 0.57 | 0.37 | 3.08e-2 | 3420.63s |
| MLF-DPO | **0.37** | 0.27 | 0.59 | 2.55e-9 | **0.89** | 0.46 | 0.60 | **0.85** | 0.44 | 0.57 | **0.79** | **0.93** | 0.49 | **0.62** | 1511.82s |
| MLR-DPO | 0.25 | 0.21 | 0.61 | 3.63e-7 | 0.81 | 0.46 | 0.60 | 0.79 | 0.44 | **0.58** | 0.69 | 0.92 | 0.49 | 0.60 | 1509.28s |
| NORM-DPO | 0.17 | **0.19** | 0.61 | **1.40e-6** | 0.77 | 0.46 | 0.60 | 0.66 | **0.45** | 0.57 | 0.58 | 0.89 | 0.48 | 0.58 | **1385.32s** |
| NPO | 0.27 | 0.11 | 0.72 | **3.36e-2** | 0.72 | 0.46 | 0.62 | 0.86 | 0.45 | 0.59 | 0.35 | 0.29 | 0.36 | 0.47 | 2407.90s |
| MLF-NPO | **0.33** | 7.25e-2 | 0.62 | 6.54e-4 | **0.94** | 0.46 | 0.62 | **0.89** | 0.45 | 0.59 | **0.63** | **0.73** | 0.47 | **0.60** | 1063.28s |
| MLR-NPO | **0.33** | **0.12** | 0.61 | 3.63e-7 | 0.91 | **0.47** | 0.62 | 0.86 | 0.45 | 0.59 | 0.61 | 0.72 | **0.47** | **0.60** | 1060.32s |
| NORM-NPO | 0.32 | 4.10e-2 | **0.63** | 9.96e-3 | 0.90 | **0.47** | **0.63** | 0.86 | 0.45 | 0.59 | 0.60 | 0.67 | **0.47** | 0.59 | **932.37s** |

