# OpenReview forum: "Structure-Aware Parameter-Efficient Machine Unlearning on Transformer Models"
_ICLR.cc/2025/Conference — Submitted to ICLR 2025_

### Official Review · Reviewer_fgpp · 2024-10-30

**Soundness:** 2
**Presentation:** 2
**Contribution:** 2
**Rating:** 3
**Confidence:** 4

**Summary:**

The authors tackle the challenge of efficient machine unlearning in large Transformer models, essential for meeting privacy regulations. Existing methods, often structure-agnostic, struggle to accurately target influence-critical parameters in Transformers. To address this, the authors introduce SPE-Unlearn, a structure-aware approach that uses learnable masks to identify key parameters within Transformer heads and filters. Optimized via a greedy search, SPE-Unlearn enhances unlearning by balancing efficiency and effectiveness. Extensive experiments show that SPE-Unlearn significantly improves unlearning performance across various Transformer models and datasets.

**Strengths:**

1. The paper is clearly written, well-formatted, and well-organized.

2. The mathematical derivations are rigorous.

**Weaknesses:**

1. The paper mentions that SPE-Unlearn can enhance the effectiveness of various methods; however, the authors only integrate SPE-Unlearn with SO and do not test it with other methods. In Table 2, SPE-SO does not show a significant improvement over SO, while requiring more time.

2. Common LLM unlearning tasks, such as TOFU, MUSE, and WMDP, are missing.

3. Several standard baselines, like NPO and "I don't know," are not included, which weakens the argument.

4. For robustness, the authors employ memory-free and memory-aided unlearning but do not explore other approaches, such as jailbreak prompts.

**Questions:**

1. Could the authors explore additional LLM unlearning benchmarks, such as TOFU, MUSE, and WMDP?

2. Could the authors evaluate more unlearning methods, like NPO and "I don't know"?

3. For robustness evaluation, would it be possible to include tests like relearning attacks or jailbreak prompts?

4. Additionally, could the authors consider using gradient norm alone to identify saliency? This might offer a more streamlined approach.

---

> ### Author Response · Authors · 2024-11-22
> **Response to Reviewer fgpp [part1]**
>
> Thank you for the constructive feedback, which has helped us improve the quality of our paper. We explain our work below:
>
> Unlearning can be broadly divided into two components: the design of the objective function and the development of the optimization algorithm. For GA, the objective function focuses solely on maximizing the loss for the unlearning data. In contrast, GD's objective function not only aims to maximize the loss on the unlearning data but also strives to maintain the performance on the remaining data. NPO targets maximizing the discrepancy in loss on the unlearning data between the original and unlearned models, while DPO minimizes the loss associated with the substitute response "I don't know" for the forget data. Fundamentally, GD, NPO and DPO all work to increase the loss on forget data, essentially serving as optimization strategies for gradient ascent.
>
> Our work primarily focuses on optimizing the design of algorithms to accelerate the process under a given objective function. The current manuscript specifically addresses second-order update methods, utilizing a second-order objective function aimed at minimizing the loss on the remaining dataset. In principle, alternative objective functions can also be employed, enabling the development of corresponding structure importance evaluation functions.The importance function for maximizing the loss on the forgetting dataset can be expressed as follows:
>
> $$
> \textnormal m^\* \approx \arg\max_{\textnormal{m}} \sum_i \Big\[ (1 - \textnormal m_i)  \big\[ \sum_{x \in \mathcal D_\mathrm r} \nabla_\textnormal m  \ell (\mathbb 1; {\mathbf\theta}^*, x)\big\] \_i+ \frac{1}{2} (1 - \textnormal m_i)^2 \big[\widehat{\mathcal I}(\mathbb 1; {\mathbf\theta}^\*, \mathcal D_\mathrm f)\big]\_i\Big]
> $$
>
> To this end, we have incorporated additional experimental scenarios by applying GA, GD, NPO, DPO and other methods on the LLama2-7B model and with TOFU benchmark, utilizing various structure importance evaluation functions. The following are the experimental results for TOFU Forget05 (due to content limitations, we include it in the next part).
>
> We uniformly applied a sparsity of 90% across these methods. **MLF** refers to a structure importance evaluation function derived from the GA objective function (i.e. **M**aximize the **L**oss on the **F**orgetting dataset). Additionally, **SPE** represents a structure importance evaluation function designed using the original objective function in second-order unlearning. Finally, **NORM** utilizes the gradient norms of the forget data to determine structure importance.
>
> Our experimental results indicate that sparse updates are better suited for unlearning than full updates, as they provide a stronger guarantee of unlearning while more effectively preserving performance. Additionally, while NORM significantly reduces computation time, SPE offers a more balanced trade-off between unlearning efficacy and model fidelity. Furthermore, structure evaluation functions with MLF also exhibit strong performance.
>
> We hope these explanation and additions to our manuscript address the concerns raised and merit a reconsideration of the score by the reviewer. In future work, we plan to include additional experiments, such as the WMDP benchmark, to further demonstrate the effectiveness of our approach.

---

> ### Author Response · Authors · 2024-11-22
> **Response to Reviewer fgpp [part2]**
>
> |  **Method**  | **Efficacy** |             |          |              |   **Fidelity**    |                 |              |                   |                 |              |                   |                 |              |              |
> | :----------: | :----------: | :---------: | :------: | :----------: | :---------------: | :-------------: | :----------: | :---------------: | :-------------: | :----------: | :---------------: | :-------------: | :----------: | :----------: |
> |              |  Forget Set  |             |          | FQ$\uparrow$ |   Real Authors    |                 |              |    World Facts    |                 |              |    Retain Set     |                 |              | MU$\uparrow$ |
> |              |   Rouge-L    |    Prob.    |    TR    |              | Rouge-L$\uparrow$ | Prob.$\uparrow$ | TR$\uparrow$ | Rouge-L$\uparrow$ | Prob.$\uparrow$ | TR$\uparrow$ | Rouge-L$\uparrow$ | Prob.$\uparrow$ | TR$\uparrow$ |              |
> |    **GA**    |     0.10     |   3.62e-3   |   0.65   | **4.31e-4**  |       0.63        |      0.35       |     0.49     |       0.85        |      0.40       |     0.53     |       0.17        |      0.02       |     0.46     |     0.11     |
> |  **MLF-GA**  |   **0.20**   | **3.93e-3** |   0.62   |   1.18e-4    |     **0.86**      |    **0.41**     |   **0.56**   |     **0.88**      |      0.41       |     0.57     |     **0.40**      |    **0.31**     |   **0.46**   |   **0.48**   |
> |  **SPE-GA**  |     0.17     |   2.31e-3   |   0.61   |   4.75e-5    |       0.83        |      0.42       |     0.58     |       0.86        |    **0.42**     |   **0.58**   |       0.33        |      0.17       |     0.45     |     0.42     |
> | **NORM-GA**  |     0.18     |   1.08e-3   |   0.56   |   1.21e-10   |       0.85        |      0.40       |     0.55     |       0.86        |      0.40       |     0.57     |       0.36        |      0.22       |     0.44     |     0.44     |
> |    **GD**    |     0.30     |   1.79e-2   |   0.54   |   2.83e-4    |       0.79        |      0.35       |     0.49     |       0.87        |      0.38       |     0.53     |       0.46        |      0.42       |   **0.50**   |     0.49     |
> |  **MLF-GD**  |   **0.37**   |  **0.15**   |   0.61   |   2.83e-4    |     **0.94**      |      0.44       |     0.59     |       0.86        |    **0.43**     |   **0.57**   |     **0.81**      |      0.91       |     0.48     |   **0.61**   |
> |  **SPE-GD**  |     0.33     |   2.31e-2   | **0.64** |   **0.63**   |       0.90        |    **0.46**     |   **0.60**   |     **0.87**      |    **0.43**     |   **0.57**   |       0.79        |      0.88       |     0.48     |   **0.61**   |
> | **NORM-GD**  |     0.37     |  **0.15**   |   0.61   |   2.83e-4    |     **0.94**      |      0.44       |     0.59     |       0.86        |    **0.43**     |   **0.57**   |     **0.81**      |    **0.91**     |     0.48     |   **0.61**   |
> |   **DPO**    |   4.57e-2    |    0.64     | **0.62** |   6.57e-12   |       0.57        |      0.46       |   **0.60**   |       0.83        |    **0.46**     |     0.57     |       0.23        |      0.73       |     0.40     |     0.47     |
> | **MLF-DPO**  |   **0.30**   |    0.19     |   0.60   |   8.06e-7    |     **0.92**      |      0.46       |     0.59     |       0.86        |      0.45       |   **0.58**   |     **0.76**      |    **0.88**     |   **0.48**   |   **0.61**   |
> | **SPE-DPO**  |     0.28     |   9.35e-2   | **0.62** | **1.84e-4**  |       0.90        |      0.46       |   **0.60**   |     **0.87**      |      0.45       |   **0.58**   |       0.69        |    **0.88**     |   **0.48**   |   **0.61**   |
> | **NORM-DPO** |     0.19     |  **0.14**   |   0.61   |   4.75e-5    |       0.88        |      0.46       |   **0.60**   |       0.82        |      0.45       |   **0.58**   |       0.66        |      0.78       |   **0.48**   |     0.60     |
> |   **NPO**    |     0.34     |    0.11     | **0.66** | **1.18e-4**  |     **0.94**      |      0.33       |     0.42     |       0.89        |      0.38       |     0.49     |       0.42        |      0.36       |     0.46     |     0.46     |
> | **MLF-NPO**  |     0.33     |    0.12     |   0.61   |   1.11e-5    |       0.91        |    **0.44**     |   **0.58**   |     **0.88**      |    **0.42**     |     0.56     |       0.74        |      0.83       |   **0.48**   |   **0.60**   |
> | **SPE-NPO**  |   **0.34**   |  **0.15**   |   0.59   |   8.11e-8    |       0.90        |    **0.44**     |   **0.58**   |       0.87        |    **0.42**     |   **0.57**   |     **0.76**      |    **0.85**     |   **0.48**   |   **0.60**   |
> | **NORM-NPO** |     0.32     |    0.12     |   0.61   |   1.11e-5    |       0.91        |      0.43       |   **0.58**   |     **0.88**      |    **0.42**     |     0.56     |       0.74        |      0.82       |   **0.48**   |     0.59     |

---

> ### Author Response · Authors · 2024-11-28
> **A Gentle Reminder of the Post-rebuttal Feedback**
>
> Thank you very much again for your initial comments. They are very valuable for improving our work. We would be grateful if you could have a look at our response and modifications and please let us know if there is anything else that can be added to our next version. We are also willing to have further discussions with you.

---

> ### Author Response · Authors · 2024-11-30
> **A Second Reminder of the Post-rebuttal Feedback**
>
> Dear Reviewer fgpp,
>
> We greatly appreciate your initial comments. We totally understand that you may be extremely busy at this time. But we still hope that you could have a quick look at our responses to your concerns. We appreciate any feedback you could give us. We also hope that you could kindly update the rating if your questions have been addressed. We are also happy to answer any additional questions before the rebuttal ends.
>
> Best Regards,
>
> Paper 14252 Author(s)

---

> ### Author Response · Authors · 2024-12-02
> **Reminder of the Post-rebuttal Feedback and Summary of Our Response**
>
> Dear Reviewer fgpp:
>
> Thank you for your time and effort in evaluating our work. We greatly appreciate your initial comments. Your insights and suggestions are extremely valuable to us.
>
> Given that we have only one day left for discussion, we are hoping to receive any additional feedback or question you might have at your earliest convenience. Your expertise would be of great help to us in improving the quality and rigor of our work.
>
> To facilitate the discussion, we would like to summarize our response as follows.
>
> - **We clarified the principles of our method**, including the rationale for using the objective function in our manuscript for second-order unlearning updates.
> - **We discussed the details of our experiments**, such as deriving an alternative objective function to identify saliency, adding new experiments with the TOFU benchmark on the LLama2-7B model, and applying state-of-the-art unlearning methods.
>
> If our responses address your concerns, we kindly request that you reconsider your evaluations. We would also be grateful for any additional comments or suggestions you might have to refine our work.
>
> Best regards,
>
> Paper 14252 Author(s)

---

### Official Review · Reviewer_Lm6c · 2024-10-31

**Soundness:** 3
**Presentation:** 4
**Contribution:** 3
**Rating:** 6
**Confidence:** 3

**Summary:**

This paper proposes a structure-aware parameter efficient machine unlearning approach for transformer models.

**Strengths:**

The research problem is interesting and the approach is novel.

**Weaknesses:**

Some experiments are insufficient, and the experimental setup should be more comprehensive.

For example, in page 7 the author notes that if the number of unlearning requests exceeds a certain threshold, the model must be retrained from scratch to regain its performance. What exactly is this threshold?

Additionally, in Table 3, is there a specific reason for selecting 90% sparsity instead of alternatives like 85% or 95%? Based on Figures 2 and 3, is this value specific to the model?

**Questions:**

1. I am interested in the comparison results with other unlearning approaches presented in Table 1. Could you either provide the results of applying the memory-aided unlearning approach to the experiments in Table 1, or explain why this comparison was not included?
2. In the experiments on successive unlearning scenarios on page 9,  the experiments shows the number of requests affect the results. How does the Volume of data affect the performance?

---

> ### Author Response · Authors · 2024-11-22
> **Response to Reviewer Lm6c**
>
> We thank the reviewer for the positive rating of our paper. We also appreciate the reviewer for acknowledging the novelty of this work and all the constructive suggestions. We hope the following clarifications can address the reviewer's concerns.
>
> >Some experiments are insufficient, and the experimental setup should be more comprehensive.
> >
> >- For example, in page 7 the author notes that if the number of unlearning requests exceeds a certain threshold, the model must be retrained from scratch to regain its performance. What exactly is this threshold?
> >- Additionally, in Table 3, is there a specific reason for selecting 90% sparsity instead of alternatives like 85% or 95%? Based on Figures 2 and 3, is this value specific to the model?
>
> Thank you for these insightful suggestions. For Question 1, the threshold for unlearning varies across different models, and there is no definitive standard. It can be determined based on specific practical requirements and scenarios. For Question 2, we investigated the sparsity for DistilBERT, BERT-base, and RoBERTa-large. We found that 90% represents a reasonable balance, achieving effective unlearning with minimal cost to model fidelity. While 85% is also feasible,  we observed a significant drop in the model's utility when meeting the unlearning criteria at 95%. Additionally, sparsity may depend not only on the model itself but also on the specific task and dataset involved. Furthermore, we have conducted further experiments, including testing the LLaMA2-7B model on the TOFU benchmark, and incorporated several state-of-the-art methods.
>
> >I am interested in the comparison results with other unlearning approaches presented in Table 1. Could you either provide the results of applying the memory-aided unlearning approach to the experiments in Table 1, or explain why this comparison was not included?
>
> We apologize for the confusion. In the context of successive unlearning, we have divided second-order unlearning (SO) into two types: memory-aided unlearning and memory-free unlearning. Our focus here is primarily on second-order unlearning. Therefore, we only provided an analysis of how full updates in SO impact memory-free unlearning. Moreover, the comparison between memory-free and memory-aided approaches is illustrated in Figure 4.
>
> >In the experiments on successive unlearning scenarios on page 9, the experiments shows the number of requests affect the results. How does the Volume of data affect the performance?
>
> We appreciate this insightful suggestion to explore the relationship between the volume of unlearning data and model utility. In our current work, we have conducted only a qualitative analysis, demonstrating that increasing the amount of unlearning data leads to greater performance degradation. In future work, we will plan to investigate this relationship quantitatively.

---

> ### Author Response · Authors · 2024-11-28
> **A Gentle Reminder of the Post-rebuttal Feedback**
>
> Thank you very much again for your initial comments. They are very valuable for improving our work. We would be grateful if you could have a look at our response and modifications and please let us know if there is anything else that can be added to our next version. We are also willing to have further discussions with you.

---

> ### Author Response · Authors · 2024-11-30
> **A Second Reminder of the Post-rebuttal Feedback**
>
> Dear Reviewer Lm6c,
>
> We greatly appreciate your initial comments. We totally understand that you may be extremely busy at this time. But we still hope that you could have a quick look at our responses to your concerns. We appreciate any feedback you could give us. We also hope that you could kindly update the rating if your questions have been addressed. We are also happy to answer any additional questions before the rebuttal ends.
>
> Best Regards,
>
> Paper 14252 Author(s)

---

> ### Author Response · Authors · 2024-12-02
> **Reminder of the Post-rebuttal Feedback and Summary of Our Response**
>
> Dear Reviewer Lm6c:
>
> Thank you for your time and effort in evaluating our work. We greatly appreciate your initial comments. Your insights and suggestions are extremely valuable to us.
>
> Given that we have only one day left for discussion, we are hoping to receive any additional feedback or question you might have at your earliest convenience. Your expertise would be of great help to us in improving the quality and rigor of our work.
>
> To facilitate the discussion, we would like to summarize our response as follows.
>
> - **We discussed the details of our experiments**, including how to choose the threshold for memory-free unlearning, why we selected 90% sparsity for our experiments, and how the volume of data affects performance.
>
> If our responses address your concerns, we kindly request that you reconsider your evaluations. We would also be grateful for any additional comments or suggestions you might have to refine our work.
>
> Best regards,
>
> Paper 14252 Author(s)

---

### Official Review · Reviewer_Jb2K · 2024-11-02

**Soundness:** 2
**Presentation:** 3
**Contribution:** 2
**Rating:** 6
**Confidence:** 3

**Summary:**

This paper proposes an efficient unlearning method with multiple benefits, such as reduced time and memory costs. The proposed method is a pruning-based unlearning approach that filters out sensitive weights to forget specific data. By converting to a differential formulation for the masking variable, the authors approximate unlearned weights through a combination of masking and the Fisher information matrix. Using a second-order Taylor expansion and the convergence assumption from LeCun et al., the method is simplified to this combination. In experiments, the authors compare their approach to prior methods and report promising performance.

**Strengths:**

I believe the strongest benefit of this work is its time and memory efficiency. Across all tables, the authors report significantly reduced costs for unlearning, which highlights a promising advancement in the field of machine unlearning.

**Weaknesses:**

However, I have several concerns:

1. It is unclear why the method is considered structure-aware. While mask variables are applied to each head in the multi-head attention block, this approach is not unique to Transformers. The authors mention as "widely-adopted unlearning methods in Transformers, e.g., fine-tuning (Golatkar et al. (2020)) and gradient difference (Liu et al. (2022); Jia et al. (2024))," but they are not specified to transformer architecture. As I understand from the manuscript, the authors suggest that their method’s efficiency makes it well-suited to large-scale Transformer models, but large scale is not a Transformer-specific attribute.


2.  Although the method is not specialized for Transformer architectures, much of the paper focuses on Transformer-specific content. Reducing this content could make the paper more concise and focused.


3. In line 190, the authors assume that " L is differentiable with respect to m", to develop Equations (5-9). However, as stated in line 160, $m$ is a binary variable. This raises concerns about whether the differentiability assumption is valid.


4. In line 181, the authors state, "we formulate the unlearning objective (1) with a learnable pair of masks for the heads and filters as a constrained optimization problem." However, objective (1) aims to retain a model with the same architecture trained only on D_r. The proposed method, which involves pruning, alters the architecture, making it misaligned with this objective.


5. The experimental section lacks details on the partitioning of D_r and D_f. This split is an essential part of the experimental setup and should be clearly specified.

6. Evaluation Concerns in Table 2 and Appendix:

6-1) The unlearning performance should be evaluated by comparing it to the performance of a retrained model, as specified by Eq (1). In specific, the MIA metric should follow this protocol, with retrained models serving as the gold standard. However, the authors set the lower values of efficacy and higher values of fidelity are better without any explanation.


6-2) In Table 1, the proposed method unlearns to a greater extent (achieving 85.94% as the lowest Unlearning Accuracy) than other methods, resulting in lower Remaining Accuracy. To ensure a fair comparison, it would be better to report Remaining Accuracy and Testing Accuracy at the point where each method reaches a common Unlearning Accuracy threshold.


6-3) All methods (FT, GD, SA) have hyperparameters that control unlearning speed, such as learning rate, but there is no discussion of these parameters in the manuscript. Without this information, it’s unclear whether the evaluations are fair.


6-4)  The authors used only D_r for Fine-Tuning (FT) and Sparsity-Aware Unlearning (SA) but used both D_r and D_f for their method. This discrepancy in dataset usage should be clarified, and comparisons with more unlearning methods that use both D_r and D_f are recommended.


6-5) Many of the unlearning methods used for comparison are outdated. It would strengthen the evaluation to include more recent works, such as:

[1] Fan, C., Liu, J., Zhang, Y., Wong, E., Wei, D., & Liu, S. (2023). Salun: Empowering machine unlearning via gradient-based weight saliency in both image classification and generation. arXiv preprint arXiv:2310.12508.

[2] Chen, M., Gao, W., Liu, G., Peng, K., & Wang, C. (2023). Boundary unlearning: Rapid forgetting of deep networks via shifting the decision boundary. In Proceedings of the IEEE/CVF Conference on Computer Vision and Pattern Recognition (pp. 7766-7775).


7. (Minor) In the second sentence of the Related Work section, "Jang et al." is cited twice in a single sentence.

**Questions:**

See above

---

> ### Author Response · Authors · 2024-11-22
> **Response to Reviewer Jb2K [part1]**
>
> We sincerely thank the reviewer for their insightful comments. We are encouraged by their recognition of our work's importance and its practical simplicity. Below, we address the comments and questions by the reviewer:
>
> >It is unclear why the method is considered structure-aware. While mask variables are applied to each head in the multi-head attention block, this approach is not unique to Transformers. The authors mention as "widely-adopted unlearning methods in Transformers, e.g., fine-tuning (Golatkar et al. (2020)) and gradient difference (Liu et al. (2022); Jia et al. (2024))," but they are not specified to transformer architecture. As I understand from the manuscript, the authors suggest that their method’s efficiency makes it well-suited to large-scale Transformer models, but large scale is not a Transformer-specific attribute.
>
> We agree with your comments. Our method is not limited to Transformers, and large scale is indeed not an attribute unique to this architecture. However, Transformers are frequently deployed in large-scale environments where unlearning becomes computationally challenging. Our approach leverages learnable masks to identify specific structures within Transformers (i.e., attention heads and filters), simplifying unlearning methods by selectively focusing on the most critical components.
>
> >In line 190, the authors assume that " $L$ is differentiable with respect to $m$", to develop Equations (5-9). However, as stated in line 160, $m$ is a binary variable. This raises concerns about whether the differentiability assumption is valid.
>
> Our method uses a pair of masks to denote attention heads and filters, which are integrated into the model's forward inference process. This integration affects the computation graph during backpropagation, allowing gradients with respect to the masks to be computed. Furthermore, existing study [1] has demonstrated the feasibility of such differentiability assumptions.
>
> [1] Kwon W, Kim S, Mahoney M W, et al. A fast post-training pruning framework for transformers[J]. Advances in Neural Information Processing Systems, 2022, 35: 24101-24116.
>
> >In line 181, the authors state, "we formulate the unlearning objective (1) with a learnable pair of masks for the heads and filters as a constrained optimization problem." However, objective (1) aims to retain a model with the same architecture trained only on D_r. The proposed method, which involves pruning, alters the architecture, making it misaligned with this objective.
>
> Our method incorporates a structure evaluation process but does not rely on pruning to simplify the model. Instead, it employs sparse updates to facilitate the unlearning process, which does not alter the architecture.
>
> >The experimental section lacks details on the partitioning of $D_r$ and $D_f$. This split is an essential part of the experimental setup and should be clearly specified.
>
> We appreciate your suggestion regarding the partitioning of $D_r$ and $D_f$. We will incorporate this information in the revised manuscript to provide a clearer and more transparent experimental settings.
>
> >Evaluation Concerns in Table 2 and Appendix
>
> Thank you for these valuable suggestions. First, we have clarified the gold standard for unlearning methods in Section 4.1 of the manuscript.  We agree that aligning the performance on the forgetting dataset with the golden standard is reasonable. Next, we will include the remaining accuracy in Table 1 to enable a fairer comparison. Additionally, we will provide detailed hyperparameter settings in the Appendix. Lastly, we have incorporated new methods and experiments into our revised work, such as NPO [2] and DPO [3] on TOFU [4] using LLama2-7B.
>
> [2] Zhang R, Lin L, Bai Y, et al. Negative preference optimization: From catastrophic collapse to effective unlearning[J]. arXiv preprint arXiv:2404.05868, 2024.
>
> [3] Rafailov R, Sharma A, Mitchell E, et al. Direct preference optimization: Your language model is secretly a reward model[J]. Advances in Neural Information Processing Systems, 2024, 36.
>
> [4] Maini P, Feng Z, Schwarzschild A, et al. Tofu: A task of fictitious unlearning for llms[J]. arXiv preprint arXiv:2401.06121, 2024.
>
> We hope our responses have adequately addressed your previous concerns. We look forward to hearing from you and would be happy to address any remaining concerns that you may still have.

---

> ### Author Response · Authors · 2024-11-22
> **Response to Reviewer Jb2K [part2]**
>
> | **Method**  | **Efficacy** |         |      |              |   **Fidelity**    |                 |              |                   |                 |              |                   |                 |              |              |
> | :---------: | :----------: | :-----: | :--: | :----------: | :---------------: | :-------------: | :----------: | :---------------: | :-------------: | :----------: | :---------------: | :-------------: | :----------: | :----------: |
> |             |  Forget Set  |         |      | FQ$\uparrow$ |   Real Authors    |                 |              |    World Facts    |                 |              |    Retain Set     |                 |              | MU$\uparrow$ |
> |             |   Rouge-L    |  Prob.  |  TR  |              | Rouge-L$\uparrow$ | Prob.$\uparrow$ | TR$\uparrow$ | Rouge-L$\uparrow$ | Prob.$\uparrow$ | TR$\uparrow$ | Rouge-L$\uparrow$ | Prob.$\uparrow$ | TR$\uparrow$ |              |
> |   **RT**    |     0.39     |  0.15   | 0.67 |     1.0      |       0.96        |      0.42       |     0.55     |       0.90        |      0.40       |     0.53     |       0.98        |      0.99       |     0.46     |     0.62     |
> |   **FT**    |     0.92     |  0.97   | 0.51 |   8.33e-16   |       0.94        |      0.47       |     0.61     |       0.89        |      0.44       |     0.57     |       0.93        |      0.96       |     0.48     |     0.63     |
> |   **SA**    |     0.97     |  0.99   | 0.51 |   3.43e-16   |       0.94        |      0.45       |     0.58     |       0.87        |      0.42       |     0.55     |       0.98        |      0.99       |     0.48     |     0.62     |
> |   **GA**    |     0.10     | 3.62e-3 | 0.65 |   4.31e-4    |       0.63        |      0.35       |     0.49     |       0.85        |      0.40       |     0.53     |       0.17        |      0.02       |     0.46     |     0.11     |
> | **SPE-GA**  |     0.17     | 2.31e-3 | 0.61 |   4.75e-5    |       0.83        |      0.42       |     0.58     |       0.86        |      0.42       |     0.58     |       0.33        |      0.17       |     0.45     |     0.42     |
> |   **GD**    |     0.30     | 1.79e-2 | 0.54 |   2.83e-4    |       0.79        |      0.35       |     0.49     |       0.87        |      0.38       |     0.53     |       0.46        |      0.42       |     0.50     |     0.49     |
> | **SPE-GD**  |     0.33     | 2.31e-2 | 0.64 |     0.63     |       0.90        |      0.46       |     0.60     |       0.87        |      0.43       |     0.57     |       0.79        |      0.88       |     0.48     |     0.61     |
> |   **DPO**   |   4.57e-2    |  0.64   | 0.62 |   6.57e-12   |       0.57        |      0.46       |     0.60     |       0.83        |      0.46       |     0.57     |       0.23        |      0.73       |     0.40     |     0.47     |
> | **SPE-DPO** |     0.28     | 9.35e-2 | 0.62 |   1.84e-4    |       0.90        |      0.46       |     0.60     |       0.87        |      0.45       |     0.58     |       0.69        |      0.88       |     0.48     |     0.61     |
> |   **NPO**   |     0.34     |  0.11   | 0.66 |   1.18e-4    |       0.94        |      0.33       |     0.42     |       0.89        |      0.38       |     0.49     |       0.42        |      0.36       |     0.46     |     0.46     |
> | **SPE-NPO** |     0.34     |  0.15   | 0.59 |   8.11e-8    |       0.90        |      0.44       |     0.58     |       0.87        |      0.42       |     0.57     |       0.76        |      0.85       |     0.48     |     0.60     |
>
> Due to space limitations, only a subset of the experimental results is presented. Our method is integrated into the current state-of-the-art unlearning approaches for large models, enabling sparse model updates.

---

> ### Comment · Reviewer_Jb2K · 2024-11-25
>
> As I understand, the authors can upload a revised draft at this time. Then, I'm confused about the review policy for addressing posted comments without submitting a new draft.

---

> > ### Author Response · Authors · 2024-11-25
> > **Response to Reviewer Jb2K**
> >
> > Thank you for your response. We upload the revised manuscript right now.

---

> > > ### Comment · Reviewer_Jb2K · 2024-11-26
> > >
> > > Thank you for your efforts in revising the manuscript.
> > >
> > > Some of my questions (such as differentiability) have been addressed, so I have slightly increased my rating.
> > >
> > > However, I still believe this paper requires further improvement.
> > >
> > > As I understand it, this work combines sparsity-based unlearning with the prior study by Kwon et al. However, I find it unclear why this work is described as structure-aware, and the manuscript contains many redundant sections.

---

> > > > ### Author Response · Authors · 2024-11-26
> > > > **Response to Reviewer Jb2K**
> > > >
> > > > We are grateful for your recognition of our work and the valuable suggestions, which we will significantly improve the quality and clarity of our paper.
> > > >
> > > > > I find it unclear why this work is described as structure-aware.
> > > >
> > > > We apologize for any confusion caused by our terminology. Our use of the term "structure-aware" is derived from the pruning literature, where the removal of certain structured groups of parameters—such as rows, columns, or blocks of a weight matrix—is referred to as structured pruning [1]. In the context of Transformers, heads and filters are critical modules, and their associated parameters can also be considered structured groups. Based on this perspective, we describe our approach as structure-aware parameter-efficient unlearning.
> > > >
> > > > [1] Wang Z, Wohlwend J, Lei T. Structured Pruning of Large Language Models[C]//Proceedings of the 2020 Conference on Empirical Methods in Natural Language Processing (EMNLP). 2020: 6151-6162.
> > > >
> > > > >The manuscript contains many redundant sections.
> > > >
> > > > We agree with your comments. Given that our work focuses on unlearning in Transformers, we acknowledge the need for some explanations of Transformers to provide context. However, we also recognize that certain sections, such as the description of Transformer in Section 2.2, are redundant. Additionally, we have reduced excessive text about Transformers in the manuscript (e.g., Multi-head Attention and Feed-Forward Network). We believe this revision improves the conciseness and focus of the manuscript.

---

### Official Review · Reviewer_ZE8b · 2024-11-02

**Soundness:** 3
**Presentation:** 2
**Contribution:** 2
**Rating:** 5
**Confidence:** 3

**Summary:**

This paper introduces a sparsity-aware machine unlearning method. First, it proposes to adaptively identify influence-critical parameters using a derived score function. Then, it applies existing machine unlearning methods (e.g., second-order unlearning) exclusively to optimize the identified parameters. The process of parameter identification and optimization is performed iteratively.

**Strengths:**

1. The approach to identifying influence-critical parameters based on the retain set (and the forget set) appears original. The derivation is clear and reasonable. (However, the necessity needs clarification, as mentioned in the weaknesses below.)
2. The paper is well-organized and easy to understand.
3. It is meaningful to explore the applications of the proposed method for both memory-free and memory-aided successive machine unlearning scenarios.

**Weaknesses:**

1. In line 273, it appears that the differentiation is performed on $\theta$ rather than $m$.
2. Doubts regarding the necessity of the proposed parameter identification approach. The authors derive a score function to identify informative parameters, involving calculating the derivative with respect to $m$ over $D_f$ and $D_r$. Why not directly optimize the mask parameters base on Eq.(4), which seems computationally cheaper and simpler with only once gradient calculation? Or why we need to transform Eq.(4) to Eq.(5)?
3. Doubts about the claimed efficiency benefits. The paper claims improved efficiency for machine unlearning, but this may not be the case:
   - Computation: Compared to vanilla second-order unlearning, the proposed method requires additional calculations involving twice the differentiation to identify informative parameters and still requires computing gradients for all parameters during the unlearning phase.
   - Memory: Although masking hides certain parameters, storing the mask and associated gradients requires extra memory.
4. Insufficient complexity analysis. A comprehensive analysis is needed, detailing the complexity of both the parameter identification and unlearning phases from perspectives like memory and computation.
5. What are the specific advantages of the proposed method for the successive machine unlearning? The descriptions (e.g., in Line 362) is  too broad and hard to understand. Detailed, clear and reasonable analysis is needed.
6. It is recommended to use the standard notation format as outlined by ICLR guidelines instead of using a uniform font throughout all equations. The correct formatting guidelines can be found on the official website.

**Questions:**

1. Why do we need a layer-wise optimization (Eq.(9)) in addition to the individual optimization (Eq.(8)) during the mask selection process?
2. How is the first item in Eq.(13) derived? It is hard to follow the method description in Sec.3.3.2.

---

> ### Author Response · Authors · 2024-11-22
> **Response to Reviewer ZE8b**
>
> We thank the reviewer for their constructive and valuable feedback which improves our work. Please see our detailed response below:
>
> >Doubts regarding the necessity of the proposed parameter identification approach. The authors derive a score function to identify informative parameters, involving calculating the derivative with respect to $m$ over $D_f$ and $D_r$. Why not directly optimize the mask parameters base on Eq.(4), which seems computationally cheaper and simpler with only once gradient calculation? Or why we need to transform Eq.(4) to Eq.(5)?
>
> We agree with your suggestion. Previous study [1] has indeed employed gradient-based methods to facilitate machine unlearning. However, such approaches are heuristic and lack a solid theoretical foundation. Our contribution lies in deriving a principled method for structure evaluation through Taylor expansion, which provides a theoretical basis for the transformation from Eq.(4) to Eq.(5).
>
> [1] Ma Z, Liu Y, Liu X, et al. Learn to forget: Machine unlearning via neuron masking[J]. IEEE Transactions on Dependable and Secure Computing, 2022, 20(4): 3194-3207.
>
> >Doubts about the claimed efficiency benefits.
>
> Indeed, while our method introduces additional computations related to the structures within Transformers, the number of structures is significantly smaller than the number of parameters in Transformers, making the storage requirement for the masks minimal. Furthermore, utilizing structural masks reduces the storage and computation of gradients for the associated parameters, representing a deliberate and reasonable trade-off.
>
> >What are the specific advantages of the proposed method for the successive machine unlearning? The descriptions (e.g., in Line 362) is too broad and hard to understand. Detailed, clear and reasonable analysis is needed.
>
> We acknowledge that the original manuscript did not clearly articulate the advantages of our method, and we will address this in a future revision. In brief, for second-order optimization, our approach offers distinct benefits: in memory-aided unlearning, it accommodates a greater number of unlearning requests without significant performance degradation; in memory-free unlearning, a single-step second-order update is applied to the original model, whereas the proposed method achieves a tighter approximation error bound, thereby preserving better performance.
>
> >Why do we need a layer-wise optimization (Eq.(9)) in addition to the individual optimization (Eq.(8)) during the mask selection process?
>
> In our approach, we initially approximate the Hessian matrix using a diagonal Fisher Information Matrix (FIM), which neglects the influence of the off-diagonal elements. To address this limitation, we employ a block-diagonal FIM during the mask selection process to better capture the interactions between different structures within the same layer.
>
> >How is the first item in Eq.(13) derived? It is hard to follow the method description in Sec.3.3.2.
>
> We apologize for the confusion. Eq.(13) is derived from the study [2]. Upon receiving an unlearning request, the model provider can utilize the data stored in memory (previously unlearned data, represented by the first term) to perform unlearning on the original model.
>
> [2] Liu J, Xue M, Lou J, et al. Muter: Machine unlearning on adversarially trained models[C]//Proceedings of the IEEE/CVF International Conference on Computer Vision. 2023: 4892-4902.
>
> We hope these responses and further additions to our manuscript address the concerns raised and merit a reconsideration of the score by the reviewer.

---

> > ### Comment · Reviewer_ZE8b · 2024-12-03
> >
> > Thanks for your response. I am still confused about some of the points mentioned above:
> >
> > 1. "Why not directly optimize the mask parameters based on Eq. (4), which seems computationally cheaper and simpler with only one gradient calculation? Or why do we need to transform Eq. (4) into Eq. (5)?" The authors argue that this transformation is more theoretical, but in my view, this transformation increases computational cost (as it involves calculating the derivative twice) and also introduces additional errors (since it is an approximation). I still don't understand the necessity of transforming Eq. (4) into Eqs. (5), (6), (7), (8), and (9).
> >
> > 2. "Doubts about the claimed efficiency benefits." My concerns are: (1) Eq. (10) still requires gradients of all model parameters, which does not reduce the computational cost compared to previous methods. (2) The mask learning process requires two gradient calculations, which also increases the computational cost compared to previous methods.

---

> > > ### Author Response · Authors · 2024-12-03
> > > **Reply to Reviewer ZE8b**
> > >
> > > We thank you for the time spent reviewing our work and for your constructive comments. We sincerely hope to have further discussions with you to see if our responses address your concerns.
> > >
> > > >"Why not directly optimize the mask parameters based on Eq. (4), which seems computationally cheaper and simpler with only one gradient calculation? Or why do we need to transform Eq. (4) into Eq. (5)?" The authors argue that this transformation is more theoretical, but in my view, this transformation increases computational cost (as it involves calculating the derivative twice) and also introduces additional errors (since it is an approximation). I still don't understand the necessity of transforming Eq. (4) into Eqs. (5), (6), (7), (8), and (9).
> > >
> > > We apologize for any confusion caused by our previous explanation. In Eq. (4), the constraint refers to the number of structures to be updated, represented by the $l_0$-norm of the mask. However, since the $l_0$-norm is non-differentiable, directly solving Equation (4) is challenging [1]. To address this, we transformed Equation (4) into Equation (5) using a Taylor expansion. While this transformation introduces additional computational overhead and approximation errors, it enables us to approximate a solution effectively. Additionally, the extra computation introduced at this stage will be offset during the unlearning phase, simultaneously reducing storage overhead.
> > >
> > > [1] Kwon W, Kim S, Mahoney M W, et al. A fast post-training pruning framework for transformers[J]. Advances in Neural Information Processing Systems, 2022, 35: 24101-24116.
> > >
> > > >"Doubts about the claimed efficiency benefits." My concerns are: (1) Eq. (10) still requires gradients of all model parameters, which does not reduce the computational cost compared to previous methods. (2) The mask learning process requires two gradient calculations, which also increases the computational cost compared to previous methods.
> > >
> > > We appreciate your observation and would like to clarify that we did not compute the gradients for all parameters. Instead, we focused on calculating the gradients of all structures (i.e., heads and filters). This approach significantly reduces computational resource requirements and storage overhead because the number of structures is substantially smaller than the total number of parameters.
> > >
> > > Using these masks, we are able to compute the gradients for only a subset of parameters, further minimizing storage demands. While we introduced the computation of structural gradients, this method effectively reduces both storage requirements and the time required for parameter updates. Moreover, as demonstrated in our latest experimental results (Tables 11, 12, and 13), this approach yields even greater efficiency and performance improvements for larger models (e.g., LLaMA-2-7b).

---

> ### Author Response · Authors · 2024-11-28
> **A Gentle Reminder of the Post-rebuttal Feedback**
>
> Thank you very much again for your initial comments. They are very valuable for improving our work. We would be grateful if you could have a look at our response and modifications and please let us know if there is anything else that can be added to our next version. We are also willing to have further discussions with you.

---

> ### Author Response · Authors · 2024-11-30
> **A Second Reminder of the Post-rebuttal Feedback**
>
> Dear Reviewer ZE8b,
>
> We greatly appreciate your initial comments. We totally understand that you may be extremely busy at this time. But we still hope that you could have a quick look at our responses to your concerns. We appreciate any feedback you could give us. We also hope that you could kindly update the rating if your questions have been addressed. We are also happy to answer any additional questions before the rebuttal ends.
>
> Best Regards,
>
> Paper 14252 Author(s)

---

> ### Author Response · Authors · 2024-12-02
> **Reminder of the Post-rebuttal Feedback and Summary of Our Response**
>
> Dear Reviewer ZE8b:
>
> Thank you for your time and effort in evaluating our work. We greatly appreciate your initial comments. Your insights and suggestions are extremely valuable to us.
>
> Given that we have only one day left for discussion, we are hoping to receive any additional feedback or question you might have at your earliest convenience. Your expertise would be of great help to us in improving the quality and rigor of our work.
>
> To facilitate the discussion, we would like to summarize our response as follows.
>
> - **We clarified the details of our method**, including the rationale behind transforming Eq. (4) to Eq. (5), the necessity of layer-wise optimization, and the derivation of the first term in Eq. (13).
> - **We discussed the advantages of our method**, highlighting its efficiency benefits and its ability to perform unlearning under two successive settings.
>
> If our responses address your concerns, we kindly request that you reconsider your evaluations. We would also be grateful for any additional comments or suggestions you might have to refine our work.
>
> Best regards,
>
> Paper 14252 Author(s)

---

### Author Response · Authors · 2024-11-25
**General Response**

We would like to thank all the reviewers for their valuable comments. We hope our responses have adequately addressed your concerns. The revised manuscript has been uploaded. We consider this an excellent opportunity to improve our work and would greatly appreciate any further feedback you may have.

---

### Meta-Review · Area_Chair_8EVF · 2024-12-15

**Metareview:**

This paper proposes a structure-aware parameter-efficient machine unlearning approach (SPE-Unlearn) for Transformer architectures. After reviewing the paper and author-reviewer discussions, I find the current version has unresolved concerns and is not ready for acceptance.

### 1. Complexity Analysis (Reviewer ZE8b)
The authors failed to address Reviewer ZE8b’s request for a quantitative complexity analysis of memory and computational efficiency. Their response lacked concrete evidence.

### 2. Implementation Details (Reviewer Lm6c)
Key implementation details remain unclear:  for example, threshold setting, sparsity setting, and data volume influence.

### 3. Generalizability and Robustness (Reviewer fgpp)
Concerns about generalizability and robustness remain unresolved:
- Benchmarks: Common benchmarks like MUSE and WMDP are not included.
- Robustness: No exploration of robustness against attacks (e.g., jailbreak prompts).
- Time Cost: SPE-SO’s higher time cost (Table 2) compared to SO is unexplained.

### Conclusion
While the approach is interesting, these unresolved issues in complexity, implementation details, and robustness make it unsuitable for acceptance without significant revision.

**Additional Comments On Reviewer Discussion:**

Some significant issues raised by the reviewers remain unresolved.

---

### Decision · Program_Chairs · 2025-01-22

Reject